# The Geometry of Projection Heads: Conditioning, Invariance, and Collapse

**Faris Chaudhry** [1]

## Abstract

We develop a geometric theory of projection heads in self-supervised learning by modeling the head as a trainable Riemannian metric on the backbone representation manifold. We show that linear heads perform implicit subspace whitening, while nonlinear heads adapt local metrics to satisfy the specific topological constraints of the loss, with head depth empirically dictating this capacity. Analyzing dimensional collapse, we prove that smooth nonlinear heads natively induce negative eigenvalues in the Hessian at collapsed equilibria, making them unstable. We empirically validate this by continuously tracking the optimization geometry during training, which reveals that smooth activations like Swish can generate explicit negative curvature to escape collapse, whereas linear and ReLU heads under continuous-time gradient flow cannot, relying instead on discrete-time optimization dynamics and BatchNorm. Finally, we geometrically characterize how metric degeneracy governs the information-invariance trade-off, explaining why the head must be discarded. Evaluated across contrastive and decorrelation-based objectives on foundation models, our results demonstrate that the projection head acts as a universal geometric buffer, decoupling the semantic backbone from the rigid, destructive constraints of the pretraining objective.

## 1. Introduction

Often, labeled data is scarce, motivating the use of self-supervised learning (SSL) in the form of pretraining to learn semantic information before supervised fine-tuning. A dominant paradigm operates on the principle of representation invariance: augmented views of the same image

(e.g., rotations or recolorations) are pulled together in embedding space. This approach, alongside non-contrastive and decorrelation-based methods, has proven remarkably robust, yet it relies on a specific, nonobvious architectural choice: the projection head. Rather than applying the loss directly to the backbone representations $z$, modern methods map $z$ through a nonlinear multi-layer perceptron (MLP) $h(\cdot)$ and apply the loss to $h(z)$. Surprisingly, this head is typically discarded after training, with the raw backbone $z$ used for downstream fine-tuning.

This 'train-with, deploy-without' strategy presents a theoretical puzzle: if the head is necessary for training, why is the representation it produces suboptimal for inference? Furthermore, why must the head be nonlinear? Intuitively, the pretraining loss demands extreme geometric distortions. Consider a downstream task classifying cat breeds by coat color. If we pretrain with a contrastive loss using color-jitter augmentations without a projection head, the network is forced to become perfectly color-invariant, severely degrading the backbone's ability to perform the downstream task. The projection head acts as a disposable preconditioner—a sacrificial set of layers that absorbs these extreme invariance demands, shielding the upstream backbone.

While prior works have framed this head as an information bottleneck that filters nuisance variables (Tishby & Zaslavsky, 2015; Tan et al., 2024; Ouyang et al., 2025) or a mechanism to prevent dimensional collapse (Jing et al., 2022; Tian et al., 2021), these explanations primarily describe what happens rather than the geometric mechanics of how the head alters the optimization landscape. Specifically, existing theories do not fully explain how the head simultaneously filters information, accelerates convergence, and escapes collapsed saddle points (Dauphin et al., 2014; Ge et al., 2015).

As a contribution to the broader program of geometric deep learning (Bronstein et al., 2021), we build upon Riemannian network analysis (Ollivier, 2015) and information geometry (Amari, 2016) to analyze the projection head as a trainable Riemannian preconditioner. It induces a pullback metric that warps the representation space, aligning loss stiffness with augmentation directions and decoupling the backbone from rigid objective constraints (which require destroying information). Specifically, we prove that while linear heads

[1]Department of Computing, Imperial College London, United Kingdom. Correspondence to: Faris Chaudhry <faris.chaudhry@outlook.com>.

*Proceedings of the 43$^{rd}$ International Conference on Machine Learning*, Seoul, South Korea. PMLR 306, 2026. Copyright 2026 by the author(s).

perform implicit subspace whitening, nonlinear heads locally adapt the metric along curved trajectories, with head depth empirically scaling the geometric capacity to compress augmentation orbits. Furthermore, we show that head curvature transforms dimensional collapse into an unstable saddle point; empirical Hessian tracking reveals that smooth activations natively inject negative curvature to escape collapse, whereas linear/ReLU heads survive only via discrete-time dynamics (Cohen et al., 2021) and BatchNorm (Santurkar et al., 2018). Finally, we demonstrate that invariance objectives force the head to induce a metric singularity. This causes Fisher information degeneracy along augmentation directions (Amari, 2016), explaining why discarding the head preserves linearly separable backbone semantics—a geometric buffering effect that also extends to explicit whitening losses (e.g., VICReg (Bardes et al., 2022)) at foundation-model scale.

## 1.1. Related Work

Self-supervised learning via similarity preservation has a long history, rooted in Siamese networks and contrastive losses designed for dimensionality reduction (Hadsell et al., 2006). Early iterations focused on pretext tasks (Doersch et al., 2015), but a paradigm shift occurred with instance discrimination (Wu et al., 2018) and the scaling of contrastive objectives like InfoNCE (van den Oord et al., 2019). This culminated in the SimCLR (Chen et al., 2020a) and MoCo (He et al., 2020) frameworks, which established that massive data augmentation and large batches of negative samples can learn representations rivaling supervised counterparts. However, these methods rely heavily on a specific architectural component: the nonlinear projection head.

The empirical necessity of the projection head was first observed by Chen et al. (2020a), who demonstrated that, while the head is essential for pretraining, the representation it produces is suboptimal for downstream tasks, a phenomenon termed the guillotine effect (Bordes et al., 2023). Prior explanations suggest the head acts as an information bottleneck, filtering out nuisance variables irrelevant to the contrastive task (Jing et al., 2022; Ouyang et al., 2025). Complementary to this, recent work has explored how the implicit bias of training algorithms induces layerwise progressive feature weighting, allowing lower layers to retain features that are entirely discarded by the deeper projection head (Xue et al., 2024). While these works describe what is lost information-theoretically or through feature skew, our work provides the geometric mechanism: we prove the head induces a singular pullback metric that annihilates information along the tangent space of the augmentation manifold.

A major theoretical puzzle emerged with non-contrastive methods like BYOL (Grill et al., 2020) and SimSiam (Chen & He, 2021), which dispense with negative samples entirely.

Standard optimization theory suggests these models should suffer from dimensional collapse. Various mechanisms have been proposed to explain why they avoid this, including BatchNorm (Richemond et al., 2020), stop-gradient operations (Chen & He, 2021), and the implicit bias of SGD in deep linear networks (Tian et al., 2021; Wen & Li, 2022). Unlike these works, which focus on first-order dynamics or linear architectures, we show that the intrinsic curvature of a smooth nonlinear head (e.g., Swish or GELU) is sufficient to natively destabilize collapsed equilibria, converting stable minima into unstable saddle points.

Our work connects the projection head to natural gradient descent (Amari, 1998) and Riemannian optimization. Explicit whitening approaches, such as Barlow Twins (Zbontar et al., 2021) and VICReg (Bardes et al., 2022), enforce isotropy by adding decorrelation terms to the loss. In contrast, we propose the standard MLP projection head performs implicit whitening. By learning a state-dependent metric, the head acts as a trainable preconditioner that warps the representation space to align with the loss Hessian (Yang & Hu, 2021). Furthermore, because the local geometry of the loss basin controls optimal choice of hyperparameters like contrastive temperature (Chaudhry, 2026), this dynamic preconditioning is essential for maintaining navigable and stable optimization trajectories (Chaudhry et al., 2026).

Finally, studies on invariance often treat augmentations as group actions. Tiwari & Konidaris (2022) explored the interaction between data geometry and augmentation, suggesting SSL recovers the underlying manifold structure. We extend this by analyzing the specific role of the projection head's Jacobian. Our result on metric degeneracy formalizes the trade-off between the invariance required by the pretraining task and the expressivity required for downstream transfer, providing a geometric foundation for the emergence of invariance discussed by Achille & Soatto (2018).

## 2. Problem Setup and Notation

Let $\mathcal{X}$ denote the input space, $\mathcal{Z} \subseteq \mathbb{R}^d$ be the representation space of the backbone, and $\mathcal{H} \subseteq \mathbb{R}^k$ be the embedding space of the projection head. Further, let $f_\theta : \mathcal{X} \to \mathcal{Z}$ be a backbone network, $h_\phi : \mathcal{Z} \to \mathcal{H}$ a projection head, and $\mathcal{D}$ denote the data-generating distribution on $\mathcal{X}$ from which training samples (i.e., images) $x$ are drawn. This pipeline is visualized in Figure 1.

Throughout this work, we use the standard differential geometry convention where $T_p\mathcal{M}$ denotes the tangent space to a manifold $\mathcal{M}$ at a point $p$ (e.g., $T_z\mathcal{Z}$ and $T_h\mathcal{H}$).

Consider a family of stochastic data augmentation operators $\mathcal{T}$ where each $t_\xi \in \mathcal{T}$ is a deterministic transformation conditioned on a random parameter vector $\xi \sim p(\xi)$. These parameters decompose into continuous components $\xi_c \in$

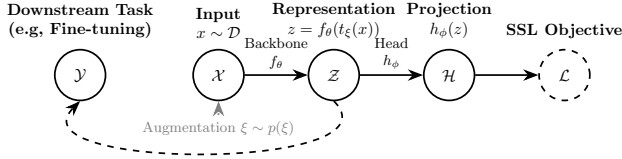

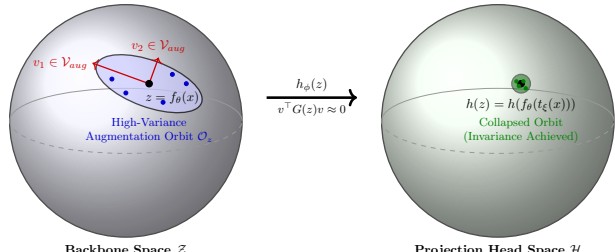

*Figure 1.* **The SSL pipeline with a projection head.** The backbone $f_\theta$ maps augmented inputs $t_\xi(x)$ to the representation manifold $\mathcal{Z}$. The projection head $h_\phi$ acts as a Riemannian preconditioner, mapping $z$ to the loss space where invariance is enforced. Supervised downstream tasks (consisting of labeled data $(x, y) \in \mathcal{X} \times \mathcal{Y}$) operate directly on $z$. The fact that we bypass $h_\phi(z)$ to use $z$ directly for the downstream task is discarding the projection head; the so-called guillotine effect.

$\mathbb{R}^m$ (e.g., color-jitter intensity or angle of rotation) and discrete components $\xi_d$ (e.g., horizontal flips). For a fixed input $x$ and fixed discrete choice $\xi_d$, continuously varying $\xi_c$ generates a smooth surface in the representation space called the augmentation orbit:

$$\mathcal{O}_z(x, \xi_d) = \{f_\theta(t_{(\xi_c, \xi_d)}(x)) \mid \xi_c \in \Xi_{\text{cont}}\},$$

which can be thought of as the space of all images generated by augmenting the original view. The goal of invariance-based learning is to approximately collapse this manifold to a single point in the projection space $\mathcal{H}$.

To analyze local invariance, we define the augmentation tangent space $\mathcal{V}_{\text{aug}}(z)$ (evaluated at the current state $z = f_\theta(t_{\xi=0}(x))$) as the subspace spanned by the infinitesimal variations of the continuous parameters:

$$\mathcal{V}_{\text{aug}}(z) = \text{span} \left\{ \frac{\partial}{\partial [\xi_c]_i} f_\theta(t_\xi(x)) \Big|_{\xi=0} \right\}_{i=1}^m \subset T_z \mathcal{Z}.$$

These tangent vectors $v \in \mathcal{V}_{\text{aug}}$ represent the local nuisance directions that the projection head must suppress. In high-dimensional representation learning, we typically assume $m \ll d$. This reflects that while the input $x$ is high-dimensional, the degrees of freedom corresponding to nuisance transformations (e.g., rotation, brightness) are relatively few. The projection head's task is to geometrically collapse these $m$ dimensions while preserving the $d - m$ semantic dimensions.

As visualized in Figure 2, the backbone representation space $\mathcal{Z}$ contains high-variance orbits spanned by $\mathcal{V}_{\text{aug}}$. The central thesis of our analysis is that the projection head acts as a trainable Riemannian metric that topologically folds this orbit into a tight equivalence class. By structurally crushing these nuisance directions to satisfy the invariance objective, the head absorbs the metric degeneracy, shielding the upstream backbone.

*Figure 2.* **The geometric role of the projection head.** In the backbone representation space $\mathcal{Z}$ (left), augmented views of an image (blue dots) are not mapped to the same point and form a high-variance orbit $\mathcal{O}_z$ spanned by the local tangent space $\mathcal{V}_{\text{aug}}$. The projection head $h_\phi$ (right) learns a local metric that compresses this orbit into a tight equivalence class, satisfying the invariance objective while shielding the upstream backbone from metric degeneracy. That is, alternative views on the same image are represented in the same way by the projection head.

### 2.1. Motivating Intuition: InfoNCE with a Linear Head

To build intuition for this geometric role, consider the standard InfoNCE loss (van den Oord et al., 2019) used in SimCLR. Suppose the projection head is strictly linear, such that $h(z) = Wz$ for some weight matrix $W \in \mathbb{R}^{k \times d}$. Ignoring normalization for clarity, the similarity between a positive pair becomes:

$$\langle h(z_i), h(z_j) \rangle = (W z_i)^\top (W z_j) = z_i^\top (W^\top W) z_j.$$

Defining $M = W^\top W$, we see that a linear projection head is equivalent to learning a Mahalanobis metric $M$ directly on the backbone representation space $\mathcal{Z}$. The optimization of $W$ is an implicit metric learning task: the head learns to warp the geometry of $\mathcal{Z}$ to maximize separation. As we will show, nonlinear heads extend this by learning a dynamic, state-dependent metric.

### 2.2. The Effective Hessian and Induced Geometry

We study the general class of pairwise SSL objectives of the form:

$$\min_{\theta, \phi} \mathbb{E}_{x \sim \mathcal{D}, \, \xi_1, \xi_2 \sim p(\xi)} \left[ \mathcal{L}(h_\phi(f_\theta(t_{\xi_1}(x))), h_\phi(f_\theta(t_{\xi_2}(x)))) \right],$$

where $\mathcal{L} : \mathcal{H} \times \mathcal{H} \to \mathbb{R}$ is the pairwise loss function.

The representation space $\mathcal{Z}$ is treated as an open subset equipped with coordinates inherited from the ambient Euclidean space. To analyze conditioning, we distinguish between the ambient dimensions of the head and the directions where the loss possesses intrinsic curvature.

**Definition 2.1** (Intrinsic Hessian Rank). The intrinsic rank $r(u, v)$ of a pairwise SSL objective at $(u, v)$ is defined as the rank of the Hessian of the loss with respect to the projection head outputs. That is, for $u = h(z)$ and $v = h(z')$,

$$r(u, v) := \text{rank}(\nabla_u^2 \mathcal{L}(u, v)) \leq k.$$

This rank determines the dimension of the transverse subspace $\mathcal{R}_u := \text{Image}(\nabla_u^2 \mathcal{L}) \subseteq T_u \mathcal{H}$, on which the loss geometry is strictly curved. Directions in the kernel $\ker(\nabla_u^2 \mathcal{L})$ correspond to invariances (e.g., radial scaling) along which the loss is locally flat. In what follows, we assume that $r(u, v)$ is constant in a neighborhood of the optimum and denote it simply by $r$.

The projection head induces a Riemannian metric structure on the representation space $\mathcal{Z}$ via the pullback of the loss geometry. To analyze the optimization dynamics, we define the effective Hessian $H_{\text{eff}}(z) \in \mathbb{R}^{d \times d}$ as the Hessian of the SSL objective with respect to the representation space. Following the chain rule, this operator decomposes into a first-order (Gauss-Newton) term and a second-order interaction term:

$$H_{\text{eff}}(z) := \underbrace{J_h(z)^\top \nabla_h^2 \mathcal{L} J_h(z)}_{\text{Pullback Metric } G(z)} + \underbrace{\sum_{i=1}^{k} [\nabla_h \mathcal{L}]_i \nabla_z^2 h_i(z)}_{\text{Interaction Term}}. \quad (1)$$

This matrix dictates the local curvature of the optimization landscape at the interface between the backbone and the projection head. While the loss Hessian $\nabla_h^2 \mathcal{L}$ resides in the $k$-dimensional embedding space, the effective Hessian $H_{\text{eff}}$ resides in the $d$-dimensional representation space and acts nontrivially only on the active pulled-back subspace $\mathcal{S}_z := \{v \in T_z \mathcal{Z} : J_h(z)v \in \mathcal{R}_{h(z)}\}$.

**Theoretical assumptions (informal).** To analyze these geometric dynamics and permit the existence of the metric tensors, we operate under a set of standard structural conditions. For readability, we summarize them informally below; the formal definitions and discussion are deferred to Appendix A. First, assume the loss and projection head are twice continuously differentiable ($C^2$), ensuring the interaction term's Hessian tensor $\nabla^2 h_\phi$ is well-defined. Furthermore, assume the optimization exhibits timescale separation, where the projection head adapts faster than the backbone (Raghu et al., 2020; Tian et al., 2021), allowing us to treat the head as a locally equilibrated preconditioner. Finally, to analyze collapse avoidance, assume that the projection head parameters are initialized generically to span indefinite matrices, that the backbone parameter-to-representation Jacobian is nondegenerate, and that architectural asymmetries (e.g., predictor lag or stop-gradients (Chen & He, 2021)) prevent the residual gradient $\|\nabla_h \mathcal{L}\|_2$ from vanishing identically near collapsed states.

## 3. Geometric Conditioning and Expressivity

The speed and stability of neural network optimization are largely dictated by the condition number $\kappa$ of the effective Hessian. If the loss landscape is highly anisotropic, gradient descent struggles with slow convergence and requires metic-

ulous learning rate tuning. In this section, we formalize how the projection head reshapes the effective loss landscape, focusing on the conditioning properties of the pullback metric $G(z)$, which dominates the optimization dynamics near stable equilibria.

### 3.1. Linear Heads: Subspace Whitening

Explicit decorrelation methods like VICReg and Barlow Twins enforce an isotropic geometry by adding covariance penalties directly to the objective. Other methods have been developed to encourage this whitening effect too (Ermolov et al., 2021). We argue that standard contrastive methods can achieve this implicitly through the projection head's pullback metric.

First consider the linear case where $h(z) = Wz$ with $W \in \mathbb{R}^{k \times d}$. While linear heads are often considered insufficient for complex tasks, they provide a baseline mechanism for resolving global anisotropy.

**Theorem 3.1** (Local Subspace Whitening). *Let $z^*$ be a fixed point. Under Assumptions 1 and 3, let $r = \text{rank}(\nabla_h^2 \mathcal{L}|_{h(z^*)})$ be the intrinsic rank of the loss. For any subspace $\mathcal{S} \subset T_{z^*}\mathcal{Z}$ of dimension $r$, there exists a linear projection head $W \in \mathbb{R}^{k \times d}$ (with $k \geq r$) such that the effective Hessian restricted to $\mathcal{S}$ is isometric to the identity on $\mathcal{S}$:*
$$v^\top H_{\text{eff}}(z^*)v = \|v\|_2^2, \quad \forall v \in \mathcal{S}.$$

Theorem 3.1 explains why linear heads may improve convergence over no head: they can whiten the spectrum of the task, such that all feature directions in the relevant subspace are learned at the same speed. However, this preconditioning is global; it applies the same distortion $W$ everywhere. Furthermore, as the rank constraint on the number of nuisance dimensions ($m \ll r$) implies, a linear head is fundamentally unable to condition directions where the loss is flat locally but curved globally. This motivates the need for nonlinear heads.

### 3.2. Nonlinear Heads: Local Metric Adaptation

In contrastive learning, the loss landscape is often curved (e.g., due to embeddings being constrained on the hypersphere from $\ell_2$ normalization or the distribution of hard negatives). A single global transformation cannot flatten such a manifold. We now show that nonlinear heads (such as MLPs) can overcome this by learning a state-dependent metric.

**Theorem 3.2** (Trajectory Linearization). *Let $\gamma(t) \subset \mathcal{Z}$ be a smooth, regular, and non-self-intersecting optimization trajectory on a compact interval $t \in [0, T]$. Under Assumptions 1 and 3, assume the intrinsic loss Hessian $\nabla_h^2 \mathcal{L}$ has constant rank $r$ along the trajectory. Then, for any $\epsilon > 0$, there exists a nonlinear projection head $h_\phi$ (parameterized*

*by an MLP) such that the induced effective Hessian is $\epsilon$-isotropic along the trajectory:*

$$\sup_{z\in\gamma(t)} \left\| P_{\mathcal{S}_z}^\top \left( J_h(z)^\top \nabla_h^2 \mathcal{L}(h(z)) J_h(z) - I_r \right) P_{\mathcal{S}_z} \right\|_F \le \epsilon.$$

This result is existential: it proves that MLPs have the capacity to condition the landscape along curved paths, a property linear heads lack.

**Proposition 3.3** (Curvature Barrier for Linear Heads). *Assume Assumption 1. Let $g_{\mathcal{L}}(u) := \nabla_u^2 \mathcal{L}(u)$ be the Riemannian metric induced by the loss. If the intrinsic geometry $(\mathcal{H}, g_{\mathcal{L}})$ has nonvanishing Riemann curvature $R_{\mathcal{L}} \not\equiv 0$, there exists no global constant linear map $W \in \mathbb{R}^{k \times d}$ such that the induced effective Hessian $H_{eff}(z) = W^\top g_{\mathcal{L}}(Wz)W$ is everywhere nondegenerate and isotropic on the optimization manifold $\mathcal{Z}$.*

It is now natural to ask what architecture the projection head requires to successfully reshape the optimization landscape. In the proof of Theorem 3.2, it is assumed the architecture of $h_\phi$ has sufficient depth and width to act as a universal approximator for the ideal metric $h^*$. In practice, restricted architectures—such as shallow or purely linear heads—incur a nonzero approximation error. The following proposition formalizes how this architectural limitation directly degrades the optimization geometry.

**Proposition 3.4** (Perturbation via Capacity Limits). *Let $h^*$ be the ideal projection head that achieves perfect isotropic conditioning, yielding a Gauss-Newton effective Hessian $H_{eff}^*$. Suppose a restricted-capacity MLP head $h_\phi$ approximates $h^*$ with a worst-case Jacobian error of $\sup_z \|J_{h_\phi}(z) - J_{h^*}(z)\|_2 \le \epsilon$. Assuming $h^*$ is L-Lipschitz ($\|J_{h^*}\|_2 \le L$) and the base loss Hessian is bounded by $\|H_{loss}\|_2 \le M$, the induced effective Hessian $H_{eff}^\phi$ satisfies:*

$$\|H_{eff}^\phi - H_{eff}^*\|_2 \le 2LM\epsilon + M\epsilon^2.$$

Consequently, while the theory requires projection heads to be sufficiently expressive approximators, it is agnostic to their specific architecture. Following the empirical ablations in foundational works like SimCLR (Chen et al., 2020a;b) and BYOL (Grill et al., 2020), the community has standardized 2- or 3-layer MLPs as the canonical architecture capable of satisfying these capacity bounds.

**Corollary 3.5** (Conditioning Failure Threshold). *For the projection head to guarantee isotropic conditioning, its approximation error must be bounded by $\epsilon < \lambda_{\min}(H_{eff}^*)/(2LM)$ (ignoring higher-order terms). If the network capacity is too shallow, $\epsilon$ exceeds this threshold, removing the conditioning guarantees and rendering the collapsed states stable.*

## 4. Stability via Curvature: Collapse Geometry

A central challenge in SSL is dimensional collapse, where the backbone maps all inputs to a constant vector or a low-rank subspace, rendering the representation useless. This is particularly puzzling in non-contrastive frameworks like BYOL and SimSiam where all samples act as positive attractors and the loss does not explicitly penalize collapse using negative samples.

We now show that the intrinsic curvature of the projection head provides a mechanical explanation for how these networks avoid this failure mode.

### 4.1. Curvature-Induced Instability

Let $z^*$ denote a collapsed configuration where $f_\theta(x) \approx c$ for all inputs. In standard MSE-based objectives, the intrinsic Hessian of the loss is PSD ($H_{\mathcal{L}} \succeq 0$). To analyze the stability of the backbone parameters $\theta$, we invoke a timescale separation argument (Assumption 7), which allows us to treat the projection head as locally equilibrated. Recalling the effective Hessian from (1):

$$H_{\text{eff}}(z^*) = \underbrace{J_h(z^*)^\top \nabla_h^2 \mathcal{L} J_h(z^*)}_{\text{Pullback Metric } G(z^*) \text{ (PSD)}} + \underbrace{\sum_{i=1}^k [\nabla_h \mathcal{L}]_i \nabla_z^2 h_i(z^*)}_{\text{Interaction Term } M(z^*)}.$$

While $G(z^*)$ is PSD, the interaction term is driven by the head's intrinsic curvature. In high-dimensional settings, the intrinsic rank $r$ of the loss is typically smaller than the representation dimension ($r < d$), and $G(z^*)$ possesses a nontrivial nullspace where the negative eigenvalues of $M(z^*)$ can emerge.

**Theorem 4.1** (Generic Instability of Collapse). *Let $z^*$ be a collapsed state. Under Assumptions 1, 6, and 7, assume the residual gradient vector $\rho = \nabla_h \mathcal{L}$ is nonzero at $z^*$.*

1. ***Linear heads preserve collapse:*** *If $h(z) = Wz$ is linear, the interaction term vanishes ($\nabla^2 h = 0$). The effective Hessian is PSD ($H_{eff} \succeq 0$). Thus, the collapsed state is a non-repelling critical region and gradient descent has no infinitesimal escape direction.*

2. ***Nonlinear heads destabilize collapse:*** *If $h_\phi$ is a nonlinear MLP with smooth activations, then under Assumption 5, the collapsed state $z^*$ is generically a strict instability region.*

**Corollary 4.2** (Instability in Parameter Space). *Under the assumptions of Theorem 4.1 and Assumption 4, if $H_{eff}(z^*)$ possesses a negative eigenvalue, then the full parameter Hessian $\nabla_\theta^2 \mathcal{L}$ also has a negative eigenvalue at $(\theta, \phi)$. Consequently, collapsed configurations are unstable saddle points for the full network objective.*

Theorem 4.1 offers a geometric resolution to the stability puzzle in non-contrastive methods. While a collapsed representation might minimize the raw objective function (e.g., zero MSE), linear heads trap the optimization in a flat valley. Conversely, smooth nonlinear heads dynamically inject negative curvature ($\lambda_{\min} < 0$), transforming the collapsed basin into a strict saddle point. Standard results in non-convex optimization (Lee et al., 2016; Sagun et al., 2018) guarantee that gradient descent escapes such strict saddle points almost surely. Thus, the projection head destabilizes trivial solutions mechanically, without requiring explicit negative samples.

### 4.2. The ReLU Gap and Discrete Dynamics

A consequence of Theorem 4.1 is that piecewise-linear activation functions (notably ReLU) lack intrinsic local instability under continuous gradient flow. Because $\nabla^2 \text{ReLU}(x) = 0$ almost everywhere, the curvature interaction term $M(z^*)$ vanishes for ReLU heads, theoretically rendering the collapsed state a non-repelling valley just like a linear head.

This theoretical ReLU gap creates a friction with empirical practice, as many successful SSL architectures apply ReLU activations. As we will demonstrate empirically, smooth activations (e.g., Swish/GELU) provide a local, native geometric guarantee of instability. In contrast, ReLU heads possess no such curvature and rely entirely on higher-order heuristics—specifically finite step-size noise (Cohen et al., 2021) and BatchNorm (Santurkar et al., 2018)—to artificially simulate escape dynamics. Our theory thus advocates for the use of smooth projection heads for robust optimization.

*Remark* 4.3 (Persistence of Instability). One might hypothesize that optimization could drive the weights into a configuration where the interaction term becomes PSD, restabilizing the collapse. However, in high-dimensional parameter spaces, the set of weight configurations yielding a PSD interaction term constitutes a measure-zero closed cone. By the transversality theorem, a generic optimization trajectory in the parameter space will intersect this measure-zero stable region only at isolated points, if at all. Furthermore, since the backbone is constantly perturbed by stochastic noise (from SGD), the system possesses a natural exploration mechanism that drives the state away from such non-repelling boundaries. Consequently, the collapsed equilibrium $z^*$ remains a strict instability region almost surely throughout the training duration, satisfying the conditions for escape established by Lee et al. (2016).

## 5. Invariance and the Information Bottleneck

The final architectural puzzle of contrastive learning is the 'train-with, deploy-without' strategy: if the projection head is essential for minimizing the loss, why is the backbone representation $z$ empirically superior for downstream tasks?

Aligning with recent information-theoretic perspectives (Ouyang et al., 2025), we provide a complementary geometric perspective. To minimize the contrastive loss, the head must mechanically destroy information to enforce invariance to nuisance variables (data augmentations). Specifically, the head must geometrically collapse the directions corresponding to augmentations, destroying information in the process. Yet, downstream tasks may require that specific information (e.g., color for image classification).

### 5.1. Linear Heads Cause Dimensional Bottlenecking

First, we consider the case of linear projection heads ($h(z) = Wz$). A pervasive empirical observation is that linear heads force the backbone representations to become lower-rank than their nonlinear counterparts. We attribute this to the interaction between the low-rank head bottleneck and the implicit bias of the optimizer. Specifically, we assume that the optimization process is biased toward minimum-norm solutions, where directions in $\mathcal{Z}$ that do not materially contribute to the loss experience an effective shrinking pressure (see Assumption 8 in Appendix A) which is standard in the analysis of implicit regularization in overparameterized regimes.

**Theorem 5.1** (Linear Rank Propagation). *Let $h(z) = Wz$ be a linear projection head with rank $k < d$. Under Assumptions 1 and 8, and assuming the backbone Jacobian is nondegenerate during training, if the loss forces the output covariance $\Sigma_h$ to be rank-deficient (rank $r \leq k$), then in the limit of training iterations, the backbone covariance $\Sigma_z$ aligns with the row space of $W$, limiting its rank to at most $k$.*

Theorem 5.1 explains why linear heads are insufficient for learning rich representations: they implicitly prune backbone dimensions that are orthogonal to the low-dimensional projection task. This is not a failure of convergence (the loss is minimized), but a failure of capacity preservation.

### 5.2. Nonlinear Heads and the Guillotine Effect

A nonlinear head relaxes the rigid global rank constraint of Theorem 5.1, allowing the backbone to remain full rank. However, to minimize the loss, the head must still satisfy local invariance requirements. We show that this produces a targeted, geometric destruction of information: the projection enforces invariance by collapsing directions tangent to augmentation orbits.

Recall the augmentation tangent space $\mathcal{V}_{\text{aug}}(z)$ defined in Section 2. Ideally, the contrastive loss forces the head to be locally invariant to these parameters such that $\nabla_\xi h(f(t_\xi(x))) \approx 0$. Assume $h$ and the augmentation map $t_\xi$ are $C^1$ in a neighborhood of $z$.

**Proposition 5.2** (Metric Singularity). *If a smooth projection head $h$ achieves local invariance to a set of continuous augmentations (i.e., the loss is minimized for all $t_\xi \in \mathcal{T}$), then the pullback metric $G(z) = J_h(z)^\top J_h(z)$ is singular. Specifically, the augmentation tangent space $\mathcal{V}_{aug}(z)$ lies in the null space of $G(z)$:*

$$v^\top G(z)v = 0, \quad \forall v \in \mathcal{V}_{aug}(z).$$

This singularity implies that the projection head acts as a geometric low-pass filter, compressing finite distances along augmentation orbits in $\mathcal{Z}$ to zero in $\mathcal{H}$. Geometrically, this corresponds to replacing $\mathcal{Z}$ with elements of the quotient space $\mathcal{Z}/\mathcal{T}$ induced by the augmentation group action, collapsing orbits to equivalence classes. While this satisfies the SSL objective, it constitutes an irreversible loss of information.

We quantify this loss using the Fisher information matrix (FIM) with respect to the augmentation parameters $\xi$.

**Theorem 5.3** (Information Hierarchy). *Let $\mathcal{I}_z(\xi)$ denote the FIM of the backbone representation $z$ with respect to augmentation parameters. If the projection head enforces invariance as per Proposition 5.2 and the augmentation manifold is nontrivial ($\dim(\mathcal{V}_{aug}) > 0$), then the information matrix of the projection is rank-deficient relative to the backbone:*

$$\mathrm{rank}(\mathcal{I}_{h(z)}) \le \mathrm{rank}(\mathcal{I}_z) - \dim(\mathcal{V}_{aug}).$$

This hierarchy geometrically formalizes the guillotine effect. Consider a downstream task that classifies cat breeds by coat color. If we pretrain with a loss that enforces invariance to color-jitter, the network must destroy color information. Theorem 5.3 proves that because the backbone lies upstream of the singular metric $G(z)$, it preserves the full dimensionality of the data manifold. Discarding the projection head is therefore not merely an empirical heuristic; it is a theoretical necessity to recover the linearly separable information filtered out during invariance learning.

## 6. Validation of Geometric Mechanisms

We empirically validate our theoretical framework, focusing on two primary mechanisms: i) curvature-induced instability and the ReLU gap (Theorem 4.1), and ii) metric singularities and orbit compression (Proposition 5.2). Full experimental details and further discussion are provided in Appendices C and D. In particular, Appendices D.1 and D.2 provide empirical validation of our theoretical assumptions. In Appendices D.3 and D.4, we confirm that these geometric effects persist across semantically denser datasets (CIFAR-100) and architectures with different inductive biases (vision transformers). Finally, we validate the architectural capacity limits with ablations on head depth (Appendix D.5).

### 6.1. Hessian Tracking and the ReLU Gap

We first aim to verify the theory of Sections 3 and 4. Namely, we track the condition number and the existence of negative eigenvalues during training from both regular and pseudo-collapsed initializations without BatchNorm to observe the geometric dynamics of escaping collapse.

As seen in Figure 3, smooth heads actively inject negative eigenvalues into the effective Hessian. During standard SimSiam training from a normal initialization (left), the condition number $\kappa$ exhibits a rapid initial spike before settling into a stable plateau, indicating that the head quickly warps the space into a highly anisotropic but consistently navigable geometry. Quantifying this relationship, we find a positive Spearman correlation ($\rho_s = 0.339$) between representation variance and condition number, supporting the argument that a healthy, expressive head natively requires a degree of local anisotropy to structure the representation space.

Further, in the case of a pseudo-collapsed initialization (middle), we observe that the first time representation variance increases is precisely when a negative eigenvalue spike occurs, forcing gradient descent away from the collapsed state. This topological phase transition triggers a violent spike in the condition number, with a Spearman correlation of $\rho_s = 0.609$ as the local basin stretches into a steep, nonconvex saddle point before dropping and gradually rising as the head dynamically reshapes the space. Notably, in the case of ReLU (right), these structural phase transitions do not occur: there is no negative curvature, and the representation variance and condition number oscillate statically around their starting values. The Spearman correlation is still high ($\rho_s = 0.669$) but instead indicates that anisotropy without a structural mechanism for escape simply locks the network in place.

We track these same dynamics from a pseudo-collapsed initialization in different conditions. As shown in Figure 4, smooth nonlinearities (left) natively destabilize the collapsed equilibrium, triggering a mechanical escape that recovers representation variance. Conversely, because ReLU and linear heads lack intrinsic continuous-time curvature, they fail to generate explicit negative curvature ($\lambda_{\min} \ge 0$). While linear heads can eventually drift out, pure ReLU networks remain fundamentally trapped under continuous-like gradient flow. They survive in practice entirely by relying on discrete-time noise and variance constraints rather than intrinsic geometric stability (right).

### 6.2. Visualizing Orbit Compression

To provide a rigorous test of the metric singularity proposed in Proposition 5.2, we analyze the high-dimensional geometry of augmentation orbits. Table 1 summarizes the geo-

**Geometric Preconditioning and Collapse Recovery**

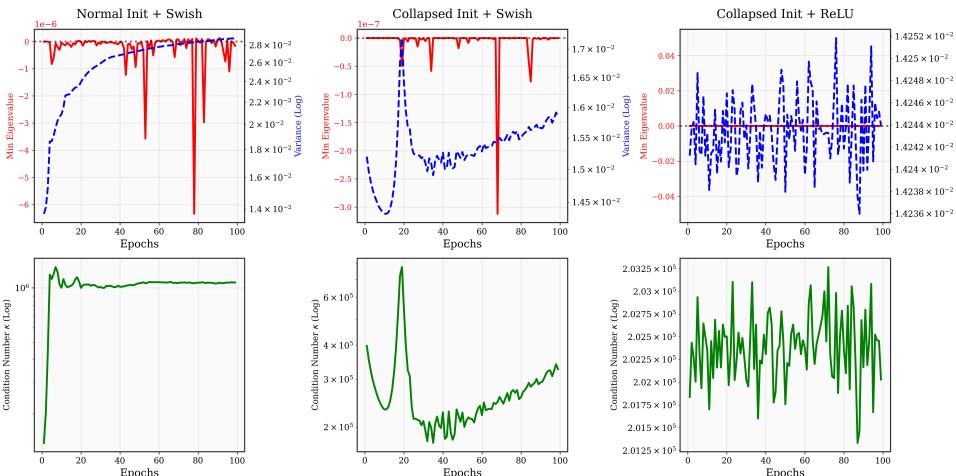

*Figure 3.* **Geometric preconditioning and collapse recovery (CIFAR-10, ResNet-18).** Smooth heads (Swish) natively inject negative curvature ($\lambda_{\min} < 0$) to navigate the landscape and aggressively escape collapsed equilibria, whereas pure ReLU networks lack this intrinsic mechanism.

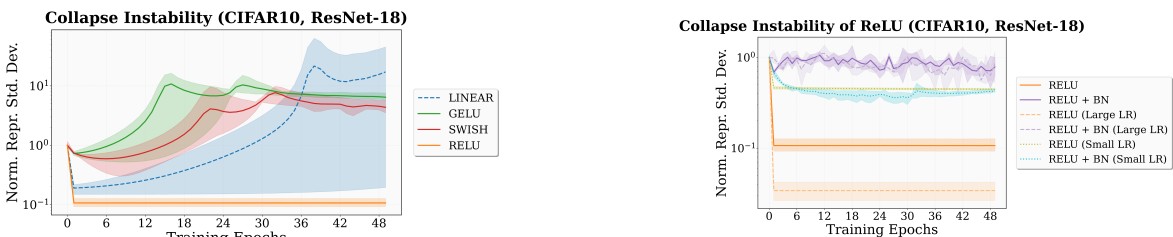

*Figure 4.* **Collapse instability and the ReLU gap.** Smooth nonlinearities (left) explicitly destabilize the collapsed equilibrium, triggering a mechanical escape that drives representation variance upward. Lacking intrinsic continuous-time curvature, ReLU networks (right) fail to generate escape directions and suffer irreversible dimensional collapse under continuous-like gradient flow (small LR, no BN). Only with BN or large LRs can variance in representation variance and thus attempt at escape be observed.

metric and information-theoretic properties of the backbone and the head.

The $21.85\times$ orbit compression provides direct empirical validation of Proposition 5.2. By reducing the mean squared spread of rotation orbits from $0.0225$ to $0.0010$, the projection head effectively annihilates variance along $\mathcal{V}_{\mathrm{aug}}$, the augmentation tangent space. This compression is not merely a byproduct of dimensionality reduction (the head is actually higher-dimensional, 2048 vs 512), but rather a targeted geometric collapse induced by the learned metric $G(z) = J_h(z)^\top J_h(z)$. As demonstrated by the higher local curvature ($1.93\times$), the head does this by geometric warping. An illustrative visualization using principal component analysis (PCA) of the orbits is shown in Figure 5.

Further, the class/orbit ratio improvement of $1.22\times$ indicates that this compression is selective: the head collapses nuisance directions (orbits shrink $4.76\times$) while relatively preserving semantic structure (classes only shrink $3.9\times$). This asymmetric compression is precisely the information-invariance trade-off formalized in Theorem 5.3. While the

head drops linear probing accuracy by 14.72%, the advantage that an MLP probe provides over a linear one nearly doubles (from 3.19% to 6.01%). This again supports the idea of the head entangling information in a nonlinear fashion rather than simply erasing it.

### 6.3. Discussion of Foundation-Scale Models

To demonstrate that these geometric phenomena are universal in SSL, we evaluated official, public foundation models across different architectures (ResNet-50, ViT-S) and loss paradigms (DINO, VICReg, Barlow Twins) that are released with the projection head. As detailed fully in Appendix D.6, our analysis reveals a fundamental geometric dichotomy driven by the pretraining objective.

When the objective requires implicit whitening (e.g., DINO's self-distillation), the projection head acts as a geometric compressor, flattening the high-dimensional augmentation manifold into a lower-dimensional subspace to force local invariance. Conversely, when the objective requires

*Table 1.* **Metrics for CIFAR-10 (ResNet-18).** Statistics are reported as mean $\pm$ standard deviation. Orbit metrics are calculated over $n = 15$ independent augmentation orbits; curvature metrics are calculated over $n = 3$ independent training seeds; probing results on the downstream task are from the final 50-epoch SimCLR representative checkpoint.

| Metric Type | Metric | Backbone ($z$) | Head ($h(z)$) | Ratio / Change |
|---|---|---|---|---|
| *Geometric* | Local Curvature | $0.125 \pm 0.002$ | $0.242 \pm 0.004$ | **1.93$\times$ Higher** |
| | Mean Orbit Spread ($\times 10^{-2}$) | $2.25 \pm 1.07$ | $0.10 \pm 0.06$ | **21.85$\times$ Collapse** |
| | Intra-orbit Distance ($D_{\text{intra}}$) | $0.211 \pm 0.045$ | $0.044 \pm 0.011$ | $4.76\times$ Reduction |
| | Inter-class Distance ($D_{\text{inter}}$) | $0.432 \pm 0.052$ | $0.111 \pm 0.014$ | $3.89\times$ Reduction |
| | Class/Orbit Ratio ($D_{\text{inter}}/D_{\text{intra}}$) | $2.04 \pm 0.52\times$ | $2.50 \pm 0.72\times$ | $1.22\times$ Separation |
| *Information* | Linear Probing Acc. | $52.27\%$ | $37.55\%$ | -14.72% Abs. Loss |
| | MLP Probing Acc. | $55.46\%$ | $43.56\%$ | -11.90% Abs. Loss |
| | **Nonlinearity Gap ($\triangle_{\text{MLP-Lin}}$)** | **+3.19%** | **+6.01%** | **+2.82% Entanglement** |

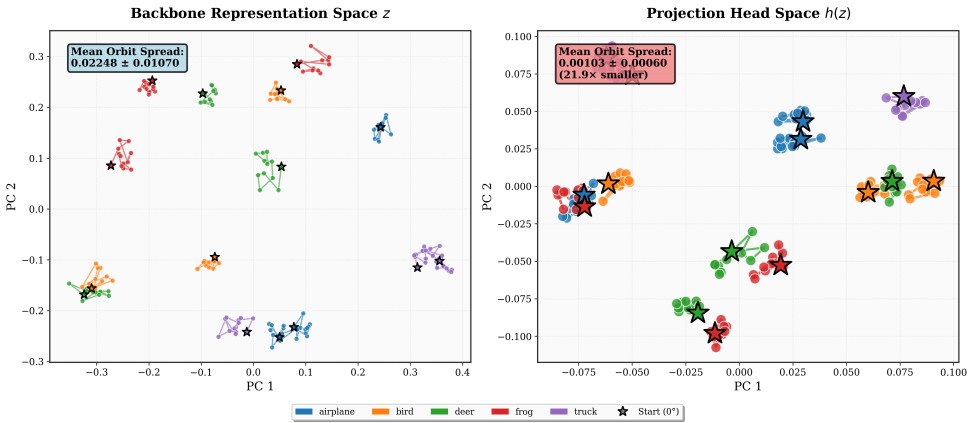

*Figure 5.* **Visualizing metric singularity (orbit collapse).** A PCA projection of augmentation orbits constructed by rotation. The star marker denotes the representation of the original, unaugmented image, serving as the anchor for each augmentation orbit. The representation space of the backbone (left) preserves the geometric variance of the rotation transformation, allowing orientation information to be linearly recovered. The learned metric (right) singularizes the space along the augmentation tangent directions, collapsing orbits into equivalence classes with $21.85\times$ less spread.

explicit whitening (e.g., VICReg's covariance penalty), it demands full-rank covariance. To satisfy this constraint without destroying the backbone's semantic clustering, the projection head acts as a dimensional expander, absorbing a local curvature jump. In all paradigms, the deep nonlinear head must act as a buffer that decouples the backbone from the destructive constraints of the objective.

## 7. Discussion and Conclusion

In this work, we developed a geometric theory of projection heads in SSL, framing them as adaptive Riemannian metrics that can whiten the optimization landscape and destabilize collapsed equilibria. Our framework provides a mechanistic explanation for the empirical necessity of projection heads, solving two central puzzles: how non-contrastive networks avoid dimensional collapse, and why heads must be discarded for downstream transfer.

Practically, replacing ReLU with smooth activations natively solves collapse via intrinsic curvature, reducing reliance on hyperparameter tuning and batch-size scaling. Furthermore, maintaining sufficient depth and nonlinearity in the projec-

tion head that equilibrates sufficiently fast is not just an empirical heuristic, but a topological requirement to shield the backbone from metric degeneracy.

**Future work.** While our results prove the existential capacity of heads to whiten the landscape, analyzing the explicit stochastic gradient dynamics required to reach these optimal configurations remains open. Additionally, extending this geometric perspective to vision transformers—where self-attention induces an in-context, data-dependent metric—presents a nontrivial extension for understanding the loss landscapes of modern architectures. Furthermore, while we define local invariance via the tangent space of continuous parameters, extending this geometric framework to discrete transformations—where the lack of a smooth group structure precludes infinitesimal modeling—remains an important frontier. Finally, our proof of metric degeneracy suggests the possibility of designing mathematically 'headless' objectives; by formulating an invertible or variance-preserving loss applied directly to the augmentation orbits, one could theoretically enforce invariance without inducing the singularity that necessitates a projection head.

## Impact Statement

This paper presents work whose goal is to advance the field of machine learning from a theoretical perspective. There are many potential societal consequences of our work, none which we feel must be specifically highlighted here.

## Software Availability

Code and results are available at the following GitHub repository.

## Acknowledgements

F.C. thanks the anonymous reviewers for their thoughtful theoretical insights and constructive feedback, which improved both the empirical presentation and overall clarity of this work.

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

# A. Discussion on Theoretical Assumptions

## A.1. List of Assumptions

We first list all theoretical assumptions. Note that Assumptions 1 and 2 are quite weak; they do not require specific assumptions about data distributions or the backbone architecture, minus the activation function used. Most geometric arguments rely on the chain rule properties of the gradient flow through $h_\phi$. Consequently, this theory broadly encompasses many contrastive (e.g., InfoNCE), non-contrastive (e.g., BYOL), and decorrelation (e.g., Barlow Twins) objectives.

**Assumption 1** (Smoothness and Curvature). *The loss function is twice continuously differentiable ($\mathcal{L} \in C^2$). We assume the projection head $h_\phi$ is a neural network equipped with smooth activation functions such that $h_\phi \in C^2(\mathcal{Z}, \mathcal{H})$.*

**Assumption 2** (Pairwise Structure). *There exists a scalar function $\ell : \mathbb{R} \to \mathbb{R}$ and a similarity or distance measure $s : \mathcal{H} \times \mathcal{H} \to \mathbb{R}$ such that*

$$\mathcal{L}(u, v) = \ell(s(u, v)).$$

While Assumptions 1 and 2 are global, we require further structural assumptions for our stability and invariance results:

**Assumption 3** (Hessian Rank and Transverse Regularity). *The intrinsic loss Hessian $\nabla_h^2 \mathcal{L}$ has constant rank $r \leq k$ in the neighborhood of the optimum. Our conditioning results apply to the $r$-dimensional transverse subspace where $\nabla_h^2 \mathcal{L}$ is positive definite.*

**Assumption 4** (Nondegenerate Backbone). *The backbone network $f_\theta$ is a local submersion at the collapsed configuration $z^*$. Equivalently, the parameter-to-representation Jacobian*

$$J_{f_\theta} : T_\theta \Theta \to T_{z^*} \mathcal{Z}$$

*is surjective (has full row rank). Thus, every tangent direction in representation space can be realized by a parameter perturbation. This assumption is generic for overparameterized MLPs at random initialization under standard schemes (e.g., Glorot, He).*

**Assumption 5** (Generic Initialization). *The projection head parameters $\phi$ are initialized from a continuous distribution (e.g., Glorot uniform). We assume the induced distribution of the Hessian tensor $\nabla^2 h(z; \phi)$ has full support on the space of symmetric matrices. This ensures the Hessian is not structurally constrained to be positive semidefinite, but rather generically spans the space of indefinite matrices.*

**Assumption 6** (Structural Asymmetry). *Assume the optimization dynamics (via predictor lag, stop-gradients, or momentum) prevent the residual gradient $\nabla_h \mathcal{L}$ from vanishing identically in the neighborhood of collapsed states. While perfect collapse ($z = z'$) implies zero gradient in MSE losses, we assume the predictor/target misalignment maintains $\|\nabla_h \mathcal{L}\|_2 > 0$ during the collapse event.*

**Assumption 7** (Timescale Separation). *Following Tian et al. (2021), we assume the projection head adapts on a faster timescale than the backbone. This assumption is supported by empirical findings in transfer learning and meta-learning, where upper layers are observed to adapt significantly faster than lower layers (Raghu et al., 2020; Fort et al., 2020). This allows us to analyze the head's geometry as being locally equilibrated to the current backbone distribution, justifying the interpretation of the head as a preconditioner.*

**Assumption 8** (Minimum-Norm Optimization). *Training minimizes a regularized objective $\mathcal{L} + \lambda \mathcal{R}(\theta)$, where $\mathcal{R}$ penalizes parameter or feature norms (e.g., via weight decay or induced feature shrinkage). Directions in $\mathcal{Z}$ that do not materially contribute to reducing $\mathcal{L}$ experience an effective shrinking pressure. This assumption is standard in analyses of implicit regularization in overparameterized regimes.*

Assumption 3 is restricted to Section 3, enabling us to use spectral whitening arguments near stable minima. Assumptions 5 and 6 are required for Section 4 to prove the instability of collapse; they ensure the existence of the negative curvature direction that destabilizes the trivial solution. Assumption 8 is used in Section 5 to derive the rank propagation and invariance results. A more detailed discussion on the validity of these assumptions follows.

## A.2. Validity of Geometric Assumptions in Practice

### A.2.1. SMOOTHNESS AND CURVATURE (ASSUMPTION 1)

Assume the projection head uses smooth activation functions (e.g., GELU, Swish, Softplus) so that the Hessian $\nabla_z^2 h$ is well defined and nontrivial. While early SSL implementations, such as the original SimCLR and BYOL, employed ReLU

activations which have zero second derivative almost everywhere, our analysis concerns the local, infinitesimal geometry of continuous-time gradient flow.

In practice, deep ReLU networks are routinely analyzed through smooth approximations (e.g., in neural tangent kernel (Jacot et al., 2018) and mean-field analyses), and many modern SSL architectures, such as vision transformers and ConvNeXt (Dosovitskiy et al., 2021; Liu et al., 2022), explicitly adopt smooth activations. For ReLU-based projection heads, the curvature-induced instability identified here does not arise through local Hessian structure, but instead through higher-order and discrete optimization effects, such as finite step sizes crossing activation kinks or BatchNorm. These effects produce effective escape directions that qualitatively mirror the negative-curvature mechanism we derive analytically for smooth networks.

We therefore emphasize that Assumption 1 isolates an idealized, analytically tractable mechanism for collapse instability; ReLU networks approximate this mechanism through discrete training dynamics or other architectural decisions rather than through local second-order curvature.

### A.2.2. GENERALITY OF PAIRWISE STRUCTURE (ASSUMPTION 2)

We assume the loss function decomposes into pairwise terms $\mathcal{L}(h(z_1), h(z_2))$. This explicitly covers:

- **Contrastive (InfoNCE):** The gradient of the log-sum-exp loss decomposes into attractive forces (positives) and repulsive forces (negatives).

- **Non-contrastive (BYOL/SimSiam):** The objective is explicitly the MSE or Cosine distance between two views.

Methods that use global statistics, such as SwAV (clustering) or DINO (centering), can be interpreted as pairwise objectives with dynamic targets (Caron et al., 2020; 2021). For example, centering acts as a dynamic bias correction on the pairwise similarity. Therefore, local rank and curvature results remain applicable to the core optimization dynamics of these methods.

### A.2.3. HESSIAN RANK AND TRANSVERSE REGULARITY (ASSUMPTION 3)

Although deep neural network objectives are globally nonconvex, our preconditioning results (Theorem 3.1) rely on the spectral properties of the loss restricted to the active subspace. Standard SSL objectives satisfy the rank regularity conditions required for our analysis:

- **InfoNCE (contrastive):** The InfoNCE loss is convex with respect to the logits. However, because the loss depends on cosine similarity, it is invariant to the radial scaling of the embeddings ($h(z) \rightarrow \alpha h(z)$). This induces a generic null direction in the intrinsic Hessian $H_{\mathcal{L}}$ corresponding to the radial ray. On the tangent space of the hypersphere (the transverse subspace), the Hessian restricted to the tangent space of the hypersphere is positive definite under mild diversity assumptions on negative samples, as empirically observed in large-batch regimes. Provided negative pairs are sufficiently diverse, $H_{\mathcal{L}}$ is positive definite on this $(k-1)$-dimensional active subspace, satisfying Assumption 3 with $r = k - 1$.

- **MSE / BYOL (non-contrastive):** The mean-squared error objective $\mathcal{L} = \|p(z_1) - \mathrm{sg}(z_2)\|_2^2$ has a Hessian locally approximated by $2J_p^\top J_p$. While $J_p$ may have a null space depending on the predictor architecture, the loss is quadratic and convex on the image of the predictor. Our analysis applies to the subspace spanned by the principal components of the predictor's Jacobian, where the curvature is strictly positive.

### A.2.4. BACKBONE REGULARITY (ASSUMPTION 4)

To link the geometric instability derived in representation space $\mathcal{Z}$ (Theorem 4.1) to the optimization of network parameters $\theta$, we rely on the nondegeneracy of the backbone mapping. This assumption ensures that the mapping from parameter space to representation space does not artificially delete directions of descent. If $H_{\text{eff}}$ has a negative eigenvalue along a direction $v \in T_{z^*}\mathcal{Z}$, surjectivity guarantees the existence of a parameter perturbation $u$ such that $J_{f_\theta} u = v$. Consequently, the negative curvature lifts to the parameter Hessian via the chain rule expansion $\nabla_\theta^2 \mathcal{L} \approx J_{f_\theta}^\top H_{\text{eff}} J_{f_\theta} + \nabla_z \mathcal{L} \cdot \nabla_\theta^2 f$. This is a generic property of overparameterized deep networks at initialization, where the number of parameters $|\theta|$ vastly exceeds the representation dimension $d$.

Indeed, this condition holds almost surely for standard feedforward networks under (i) random initialization from continuous distributions (e.g., Glorot, He), and (ii) nondegenerate nonlinearities (e.g., GELU, LeakyReLU, SoftPlus). See Sagun et al. (2018); Arora et al. (2019) for related statements on rank genericity in overparameterized regimes. This assumption fails only on a set of measure zero corresponding to degenerate critical points where neuron activations saturate or gradients vanish simultaneously; such cases are not representative of practical training dynamics.

### A.2.5. GENERIC INITIALIZATION AND NONDEGENERATE CURVATURE (ASSUMPTION 5)

Assumption 5 requires that the projection head parameters are initialized from a continuous distribution. In typical SSL architectures, heads are MLPs initialized using Gaussian or uniform schemes (e.g., Xavier/Kaiming). We assume the induced distribution of the Hessian tensor $\nabla_z^2 h(z)$ has full support on the space of symmetric matrices. This is a standard property of random MLP initialization: the second-derivative tensor at initialization is a random variable that is not structurally constrained to be PSD (as it would be in a convex network or linear network). This means that the interaction term in the effective Hessian can generically access negative eigenvalues, driving the instability mechanism.

Furthermore, while the pullback term $J_h^\top \nabla_h^2 \mathcal{L} J_h$ is PSD, it is typically rank-deficient at collapse. Under Assumption 5, the interaction term $\sum_{i=1}^k \rho_i \nabla_z^2 h_i$ has a distribution with nonzero projection onto indefinite directions in the space of symmetric matrices, ensuring that the total Hessian $H_{\text{eff}}$ inherits negative eigenvalues from the indefinite component almost surely in high dimensions.

Thus, the generic initialization assumption is not a restrictive modeling choice, but rather a weak regularity condition reflecting the almost-sure geometric properties of randomly initialized neural networks. It only rules out pathological configurations where curvature tensors vanish identically, which are irrelevant in both theory and practical training.

### A.2.6. RESIDUAL GRADIENTS AT COLLAPSE (ASSUMPTION 6)

The instability analysis in Theorem 4.1 requires that the residual gradient $\nabla_h \mathcal{L}$ does not vanish at representations arbitrarily close to collapse. This guarantees that the curvature interaction term

$$\nabla_h \mathcal{L} \cdot \nabla^2 h$$

is active, generating negative curvature directions that destabilize the collapsed solution. In modern SSL architectures, this nonvanishing gradient condition is structurally enforced via explicit architectural asymmetry:

- **BYOL & momentum encoders:** In BYOL-style frameworks, the target network parameters $\omega$ are updated via an exponential moving average (EMA) of the online network parameters $\theta$. During training, the lag induced by EMA means that $\omega \neq \theta$ except at convergence. As a result, even when backbone representations approach collapse, the target projection $h_\omega(z)$ differs from the online projection $h_\theta(z)$, producing a nonzero prediction error and therefore a residual gradient $\nabla_\theta \mathcal{L} \neq 0$.

- **SimSiam & predictor asymmetry:** In SimSiam-style methods, symmetry is broken by introducing a predictor network on only one branch together with a stop-gradient on the other. At initialization and throughout training, the predictor $p(\cdot)$ almost surely does not represent the identity map exactly. Thus, even when $z_1 = z_2$ (collapsed representations), the prediction error $\|p(z_1) - z_2\|_2$ remains nonzero, producing a persistent residual gradient signal.

A potential concern is that the predictor network $p(\cdot)$ might converge to the identity mapping (i.e., $p(z) \approx z$) faster than the backbone collapses, causing the residual gradient $\rho$ to vanish. However, in practical SSL with strong data augmentations, the target $z'$ is a stochastic view of the input. Even if the predictor is optimal, it can only learn the conditional expectation of the target view given the online view. Due to the information loss inherent in augmentation (e.g., cropping), the prediction error (and thus the residual $\rho$) is theoretically lower-bounded by the irreducible variance of the augmentation distribution. Thus, strict convergence to $\rho = 0$ is infeasible in the stochastic regime and the curvature interaction term remains active. This argument rules out vanishing residuals except at measure-zero coincidences of augmentation realizations and up to numerical precision more generally.

### A.2.7. TIMESCALE SEPARATION (ASSUMPTION 7)

This assumption is used in our stability analysis to justify treating the head Jacobian $J_h(z)$ and curvature $\nabla_z^2 h(z)$ as locally equilibrated when analyzing the instantaneous Hessian seen by the backbone. While standard training updates both

simultaneously, this effective separation is justified by the vanishing gradient tendency where upper layers receive larger gradient magnitudes. Recent studies on the loss landscape of deep networks (Fort et al., 2020) and the rapid adaptation of heads in meta-learning (Raghu et al., 2020) provide empirical support for this separation, showing that head parameters move significantly further and faster in the optimization landscape than backbone features during critical phases of training. We also verify this empirically for SSL in Appendix D.1.

Intuitively, this effective separation is justified by three factors:

1. **Gradient magnitude:** The projection head is topologically closer to the loss function. In standard backpropagation, gradients undergo attenuation as they pass through layers, a manifestation of the vanishing gradient phenomenon (Bengio et al., 1994). Consequently, the weights of the head $\phi$ typically receive larger gradient updates than the deep backbone weights $\theta$, leading to faster local adaptation. This effect may be partially mitigated in architectures with residual or skip connections, which improve gradient propagation across depth (He et al., 2016).

2. **Parameter complexity:** The projection head (typically a 2-3 layer MLP) has orders of magnitude fewer degrees of freedom than the backbone (e.g., ResNet-50 or ViT). Optimization theory suggests that lower-dimensional systems converge faster than high-dimensional ones. Thus, the head can rapidly minimize the loss relative to the current representation $z$, effectively tracking the backbone's slow drift.

3. **Empirical evidence:** The success of linear probing (where a head is trained on a frozen backbone) demonstrates that heads can converge rapidly even when the backbone is static. In the joint training limit, this implies the head effectively solves the local optimization problem at each step of the backbone's global trajectory.

### A.2.8. MINIMUM-NORM ASSUMPTION 8)

The assumption of minimum-norm bias is satisfied in virtually all practical SSL settings through several mechanisms. First, explicit regularization in standard frameworks (e.g., SimCLR, BYOL) via weight decay ($\ell_2$ penalty) imposes a persistent contractive force, driving feature parameters in the head's nullspace to zero since the loss gradient is zero in those directions. Second, even without explicit regularization, SGD on overparameterized networks exhibits an implicit bias toward minimum-norm solutions (Soudry et al., 2018; Gunasekar et al., 2018). Third, BatchNorm combined with weight decay on the affine scale parameters $\gamma$ will drive nullspace features to collapse (rank loss), preventing BatchNorm from maintaining variance in useless directions.

## B. Proofs of Results

### B.1. Proof of Theorem 3.1 (Local Subspace Whitening)

*Proof.* Let $H = \nabla_h^2 \mathcal{L}|_{h(z^*)}$ be the $k \times k$ intrinsic loss Hessian. By Assumption 3, $H$ has $r$ strictly positive eigenvalues. We perform the eigendecomposition $HU_r = U_r \Lambda_r$, where $\Lambda_r \in \mathbb{R}^{r \times r}$ is the diagonal matrix of positive eigenvalues and $U_r \in \mathbb{R}^{k \times r}$ contains the corresponding eigenvectors.

We construct the head explicitly as $W = U_r \Lambda_r^{-1/2} Q$, where $Q \in \mathbb{R}^{r \times d}$ is a semi-orthogonal matrix ($QQ^\top = I_r$) whose row spans the target subspace $\mathcal{S} \subset T_{z^*}\mathcal{Z}$. Substituting this into the effective Hessian definition for a linear head ($J_h = W$):

$$H_{\text{eff}}(z^*) = W^\top H W$$
$$= (Q^\top \Lambda_r^{-1/2} U_r^\top) H (U_r \Lambda_r^{-1/2} Q).$$

Using the eigenvector property $HU_r = U_r \Lambda_r$ and orthogonality $U_r^\top U_r = I_r$, the core term simplifies to $\Lambda_r^{-1/2} \Lambda_r \Lambda_r^{-1/2} = I_r$. Thus, the $d \times d$ effective Hessian becomes $H_{\text{eff}}(z^*) = Q^\top I_r Q$. Restricted to any vector $v \in \mathcal{S}$ in the row space of $Q$, we have $Qv \in \mathbb{R}^r$ with $\|Qv\| = \|v\|$. Therefore:

$$v^\top H_{\text{eff}}(z^*)v = v^\top Q^\top Qv = \|Qv\|_2^2 = \|v\|_2^2,$$

proving that the effective Hessian restricted to $\mathcal{S}$ is isometric to the identity. $\square$

*Remark* B.1 (Local Convexity). While deep representation learning objectives are globally nonconvex, Theorem 3.1 and the subsequent conditioning results concern the local geometry of optimization trajectories or basins of attraction. In

such regimes, the projected Hessian is PSD on the active subspace $\mathcal{S}$. This holds for standard self-supervised losses: the log-sum-exp structure of InfoNCE is convex in the logits, and squared-error objectives (e.g., BYOL, SimSiam) are quadratic. Our whitening construction relies on Assumption 3 to guarantee the existence of the real matrix square root $\Lambda^{-1/2}$.

## B.2. Proof of Theorem 3.2 (Trajectory Linearization)

*Proof.* Let $\gamma : [0, T] \rightarrow \mathcal{Z}$ be a smooth regular trajectory and define the $d \times d$ pullback metric $G(z) = J_h(z)^\top \nabla_h^2 \mathcal{L}(h(z)) J_h(z)$ (Lee, 2018). By Assumption 3, $\mathrm{rank}(G(z)) = r$ for all $z \in \gamma([0, T])$. Let $\mathcal{S}_z \subset T_z \mathcal{Z}$ be the span of the eigenvectors associated with the $r$ nonzero eigenvalues of $G(z)$. Denote by $P_{\mathcal{S}_z}$ the orthogonal projection onto $\mathcal{S}_z$.

**Smooth subbundle and induced metric.** By Assumption 1, $\mathcal{L}$ is $C^2$, so $G(z)$ is continuous. Because the rank is constant on the compact set $\gamma([0, T])$, the nonzero eigenvalues are uniformly separated from zero. By Rellich's theorem for symmetric matrices (Kato, 1995), the spectral projector $z \mapsto P_{\mathcal{S}_z}$ is smooth, and thus the assignment $z \mapsto \mathcal{S}_z$ defines a smooth rank-$r$ subbundle of $T\mathcal{Z}|_\gamma$.

We define a bilinear form $g_z$ on $\mathcal{S}_z$ by $g_z(v, w) = v^\top G(z) w$. This defines a smoothly varying inner product. We extend $g_z$ to a full Riemannian metric $\tilde{g}$ on $T\mathcal{Z}$ by choosing a smooth inner product on the orthogonal complement $\mathcal{S}_z^\perp$. Since $\gamma$ is a regular smooth curve, the tubular neighborhood theorem guarantees a neighborhood $U$ of $\gamma([0, T])$ and a coordinate chart $\psi : U \rightarrow \mathbb{R}^d$ such that, in these coordinates, the metric tensor of $\tilde{g}$ satisfies $\tilde{g}_{ij}(z) = \delta_{ij}$ for $1 \leq i, j \leq r$ (Lee, 2012).

**Ideal map and exact whitening.** Define the ideal projection head $h^* : U \rightarrow \mathbb{R}^k$ ($k \geq r$) by taking the first $r$ coordinate functions of $\psi$ and padding with zeros. Along the trajectory $\gamma$, the Jacobian $J_{h^*}(z)$ restricts to an isometry from $(\mathcal{S}_z, g_z)$ to $(\mathbb{R}^r, \langle \cdot, \cdot \rangle)$. By the definition of $g_z$ and the isometry property, for all $v, w \in \mathcal{S}_z$:

$$v^\top G(z) w = g_z(v, w) = \langle J_{h^*}(z) v, \ J_{h^*}(z) w \rangle = v^\top J_{h^*}(z)^\top J_{h^*}(z) w.$$

This implies the perfect whitening condition on the active subspace:

$$P_{\mathcal{S}_z}^\top \left( J_{h^*}(z)^\top \nabla_h^2 \mathcal{L}(h(z)) J_{h^*}(z) - I_r \right) P_{\mathcal{S}_z} = 0, \qquad \forall z \in \gamma([0, T]).$$

**Neural approximation and error control.** The set $K = \gamma([0, T])$ is compact. By the $C^1$-universal approximation theorem (Hornik, 1991; Pinkus, 1999), for any $\delta > 0$ there exists an MLP $h_\phi$ with smooth activations such that $\sup_{z \in K} \|h_\phi(z) - h^*(z)\| < \delta$ and $\sup_{z \in K} \|J_{h_\phi}(z) - J_{h^*}(z)\|_F < \delta$.

Let $H(u) = \nabla_u^2 \mathcal{L}(u)$ denote the $k \times k$ loss Hessian. On the compact domain containing the trajectory, let $M = \sup_z \|H(h_\phi(z))\|_F$ and let $L_H$ be the Lipschitz constant of $H$. We decompose the error $\Delta(z)$ between the induced effective Hessian and the identity $I_r$ on the active subspace. Writing $E(z) = J_{h_\phi}(z) - J_{h^*}(z)$ and $C_J = \sup_{z \in K} \|J_{h^*}(z)\|_F$, we expand the term via the triangle inequality:

$$
\begin{aligned}
\Delta(z) &= \|P_{\mathcal{S}_z}^\top (J_{h_\phi}^\top H(h_\phi) J_{h_\phi} - J_{h^*}^\top H(h^*) J_{h^*}) P_{\mathcal{S}_z}\|_F \\
&\leq \|J_{h^*}^\top (H(h_\phi) - H(h^*)) J_{h^*}\|_F + \|E^\top H(h_\phi) J_{h^*} + J_{h^*}^\top H(h_\phi) E + E^\top H(h_\phi) E\|_F \\
&\leq C_J^2 L_H \delta + 2\delta M C_J + M \delta^2.
\end{aligned}
$$

Because $\Delta(z) = \mathcal{O}(\delta)$, for any $\epsilon > 0$, we can choose a sufficiently small $\delta$ (realized by a finite MLP width) such that $\Delta(z) \leq \epsilon$ for all $z \in K$, satisfying the isotropic condition along the trajectory. $\square$

## B.3. Proof of Proposition 3.3 (Curvature Barrier for Linear Heads)

*Proof.* The condition for global trajectory whitening (Theorem 3.2) requires the effective Hessian $H_{\text{eff}}(z)$ to be isotropic at every point $z \in \mathcal{Z}$. In geometric terms, this requires the induced pullback metric $G(z)$ to be globally flat ($R_G \equiv 0$) and nondegenerate (as the second-order interaction term of $H_{\text{eff}}(z)$ vanishes in this linear case).

Consider a linear projection head $h(z) = Wz$, for which the Jacobian is the constant matrix $J_h = W$. If $W$ is rank-deficient on the tangent space $T_z \mathcal{Z}$, then $G(z)$ is degenerate and fails to define a Riemannian metric; in this case, the effective Hessian lacks a learning signal in the null space of $W$, failing the whitening condition by definition. We therefore restrict attention to the case where $W$ is a linear immersion.

In this case, the image $W(\mathcal{Z})$ is an immersed affine submanifold of $\mathcal{H}$. The metric $G = W^* g_{\mathcal{L}}$ coincides with the Riemannian metric induced on $W(\mathcal{Z})$ by the ambient loss metric $g_{\mathcal{L}}$. Because $W$ is a linear map, the second fundamental form of this immersion vanishes identically. By the Gauss equation, the Riemann curvature tensor of the pullback metric $R_G$ is given simply by the restriction of the ambient curvature tensor $R_{\mathcal{L}}$ to the tangent vectors in the image $W(\mathcal{Z})$.

By assumption, the loss manifold $(\mathcal{H}, g_{\mathcal{L}})$ has nonvanishing curvature ($R_{\mathcal{L}} \not\equiv 0$), such as the constant positive curvature of a hypersphere. Since the ambient curvature tensor $R_{\mathcal{L}}$ does not vanish identically, its restriction to any nondegenerate immersed affine submanifold $W(\mathcal{Z})$ cannot vanish identically. Consequently, the pullback metric $G$ necessarily inherits nonzero curvature ($R_G \not\equiv 0$). However, because a globally Euclidean metric requires $R_G \equiv 0$ everywhere, it follows that no constant linear head can satisfy the global whitening condition. Achieving global flattening of a curved landscape necessitates a state-dependent Jacobian $J_h(z)$ that can locally adapt the coordinate frame, a property unique to nonlinear heads. $\qquad\square$

### B.4. Proof of Proposition 3.4 (Perturbation via Capacity Limits)

*Proof.* Let $J^* \equiv J_{h^*}(z)$ and $J_\phi \equiv J_{h_\phi}(z)$. The approximation error is given by the error matrix $E = J_\phi - J^*$, where by assumption, the spectral norm is bounded by $\|E\|_2 \leq \epsilon$. Thus, we can write the approximate Jacobian as $J_\phi = J^* + E$.

Under the Gauss-Newton approximation of the optimization landscape, the effective Hessian with respect to the backbone representation $z$ is driven by the pullback of the loss Hessian $H_{\text{loss}}$ through the head's Jacobian. For the ideal head, this is $H_{\text{eff}}^* = (J^*)^\top H_{\text{loss}} J^*$. For the approximate head, this is $H_{\text{eff}}^\phi = J_\phi^\top H_{\text{loss}} J_\phi$.

Substituting $J_\phi = J^* + E$ into the expression for $H_{\text{eff}}^\phi$:

$$
\begin{aligned}
H_{\text{eff}}^\phi &= (J^* + E)^\top H_{\text{loss}} (J^* + E) \\
&= (J^*)^\top H_{\text{loss}} J^* + (J^*)^\top H_{\text{loss}} E + E^\top H_{\text{loss}} J^* + E^\top H_{\text{loss}} E \\
&= H_{\text{eff}}^* + (J^*)^\top H_{\text{loss}} E + E^\top H_{\text{loss}} J^* + E^\top H_{\text{loss}} E.
\end{aligned}
$$

We are interested in the spectral perturbation $\|H_{\text{eff}}^\phi - H_{\text{eff}}^*\|_2$. Subtracting $H_{\text{eff}}^*$ from both sides and applying the triangle inequality yields:

$$
\|H_{\text{eff}}^\phi - H_{\text{eff}}^*\|_2 \leq \|(J^*)^\top H_{\text{loss}} E\|_2 + \|E^\top H_{\text{loss}} J^*\|_2 + \|E^\top H_{\text{loss}} E\|_2.
$$

Since the spectral norm is submultiplicative ($\|AB\|_2 \leq \|A\|_2 \|B\|_2$) and symmetric ($\|A^\top\|_2 = \|A\|_2$), we can bound each term individually:

$$
\begin{aligned}
\|(J^*)^\top H_{\text{loss}} E\|_2 &\leq \|J^*\|_2 \|H_{\text{loss}}\|_2 \|E\|_2 \leq LM\epsilon \\
\|E^\top H_{\text{loss}} J^*\|_2 &\leq \|E\|_2 \|H_{\text{loss}}\|_2 \|J^*\|_2 \leq \epsilon ML = LM\epsilon \\
\|E^\top H_{\text{loss}} E\|_2 &\leq \|E\|_2 \|H_{\text{loss}}\|_2 \|E\|_2 \leq M\epsilon^2.
\end{aligned}
$$

Summing these bounds provides the strict upper bound on the effective Hessian perturbation:

$$
\|H_{\text{eff}}^\phi - H_{\text{eff}}^*\|_2 \leq 2LM\epsilon + M\epsilon^2.
$$

For a sufficiently expressive architecture where the approximation error $\epsilon$ is small, the $\epsilon^2$ term vanishes rapidly, yielding the first-order perturbation bound $2LM\epsilon$. However, if the network lacks sufficient depth to approximate $h^*$, the error $\epsilon$ grows large, and this perturbation actively degrades the lowest eigenvalues of $H_{\text{eff}}^*$, preventing the isotropic conditioning necessary to prevent dimensional collapse. $\qquad\square$

### B.5. Proof of Corollary 3.5 (Conditioning Failure Threshold)

*Proof.* The effective Hessians $H_{\text{eff}}^*$ and $H_{\text{eff}}^\phi$ are real symmetric matrices. By Weyl's Inequality for Hermitian matrices (Stewart & Sun, 1990), the spectral perturbation of their eigenvalues is strictly bounded by the spectral norm of their difference. Therefore, the smallest eigenvalue of the approximated Hessian is bounded by:

$$
\lambda_{\min}(H_{\text{eff}}^\phi) \geq \lambda_{\min}(H_{\text{eff}}^*) - \|H_{\text{eff}}^\phi - H_{\text{eff}}^*\|_2.
$$

Substituting the spectral perturbation bound from Proposition 3.4, we obtain:

$$\lambda_{\min}(H_{\text{eff}}^{\phi}) \geq \lambda_{\min}(H_{\text{eff}}^{*}) - (2LM\epsilon + M\epsilon^2).$$

To maintain the geometric guarantee of isotropic conditioning (and thus prevent flat or negative curvature directions from flattening the landscape), we require $\lambda_{\min}(H_{\text{eff}}^{\phi}) > 0$.

Assuming $\epsilon$ is small such that the first-order term dominates, this strictly requires:

$$2LM\epsilon < \lambda_{\min}(H_{\text{eff}}^{*}) \implies \epsilon < \frac{\lambda_{\min}(H_{\text{eff}}^{*})}{2LM}.$$

If the projection head architecture is too shallow or lacks the nonlinear capacity to adapt the metric (for example, a linear head attempting to fit a highly curved $h^*$), the Jacobian approximation error $\epsilon$ becomes large. When $2LM\epsilon \geq \lambda_{\min}(H_{\text{eff}}^{*})$, the lower bound drops below zero. In this regime, the approximation error dominates the spectrum, the condition number diverges, and the theoretical guarantees of collapse avoidance are voided. $\square$

### B.6. Proof of Theorem 4.1 (Instability of Collapse)

*Proof.* We consider the full effective Hessian $H_{\text{eff}}(z^*) = G(z^*) + M(z^*)$ defined in (1), analyzing the quadratic form $v^\top H_{\text{eff}}(z^*)v$ along a direction $v$. The first term, the pullback metric $G(z^*) = J_h^\top \nabla_h^2 \mathcal{L} J_h$, is PSD. By Definition 2.1, $\text{rank}(\nabla_h^2 \mathcal{L}) = r \leq k$. In high-dimensional representation learning, we typically have $r < d$. Consequently, the $d \times d$ matrix $G(z^*)$ is singular and possesses a nontrivial nullspace $\ker(G)$ of dimension at least $d - r$.

The stability is thus determined by the interaction term:

$$Q_{\text{int}}(v) = v^\top \left( \sum_{i=1}^{k} \rho_i \nabla^2 h_i(z^*) \right) v,$$

where $\rho = \nabla_h \mathcal{L}$ is the residual gradient vector (nonzero by Assumption 6).

**Case 1: Linear Head.** If $h(z) = Wz$, then $\nabla^2 h_k(z) = 0$ identically. Thus $Q_{\text{int}}(v) = 0$. The total Hessian is dominated by the PSD Pullback term. Thus, $H_{\text{eff}} \succeq 0$. Since there are no negative eigenvalues, there is no direction of strict descent (to second order), making the state non-repelling.

**Case 2: Nonlinear Head.** Under Assumption 5, the interaction matrix $M(\phi) = \sum_k \rho_k \nabla^2 h_k(z^*; \phi)$ is a random element of the space of symmetric matrices $\mathbb{S}^d$ with full support. The total effective Hessian is $H_{\text{eff}} = Q_{GN} + M(\phi)$.

We observe that the Gauss-Newton term $Q_{GN}$ is derived from the intrinsic loss Hessian of rank $r < d$ (Definition 2.1), and is therefore singular. Consequently, $Q_{GN}$ possesses a nontrivial nullspace where the stability of the equilibrium depends entirely on the indefinite interaction term $M(\phi)$. In the space of symmetric matrices, the set of matrices possessing at least one negative eigenvalue is open and dense.

Because the distribution of $M(\phi)$ has full support and $Q_{GN}$ is rank-deficient, the configuration of parameters $\phi$ required to keep $H_{\text{eff}}$ PSD (thereby stabilizing collapse) belongs to a proper subset of the parameter space. While this subset has nonempty interior in finite dimensions, its relative volume vanishes as the representation dimension $d$ increases, rendering the local instability generic. $\square$

*Remark* B.2 (Motivation from Random Matrix Theory). The destabilizing mechanism is motivated by Wigner's semicircle law (Rudelson & Vershynin, 2010). For a random symmetric matrix $M \in \mathbb{S}^d$, the eigenvalues are distributed over an interval proportional to $[-\sqrt{d}, \sqrt{d}]$. As $d$ increases, the spectral radius of the indefinite interaction term grows as $O(\sqrt{d})$, eventually overpowering the fixed positive eigenvalues of the Gauss-Newton term $Q_{GN}$. Even for moderate dimensions, $H_{\text{eff}}$ is overwhelmingly likely to possess negative eigenvalues, providing the infinitesimal escape directions necessary for the backbone to avoid dimensional collapse.

### B.7. Proof of Corollary 4.2 (Instability in Parameter Space)

*Proof.* We link representation-space instability to the full network parameters $\theta$. The full parameter Hessian $\nabla_\theta^2 \mathcal{L}$ follows the chain rule expansion:

$$\nabla_\theta^2 \mathcal{L} = J_{f_\theta}^\top H_{\text{eff}} J_{f_\theta} + \nabla_z \mathcal{L} \cdot \nabla_\theta^2 f.$$

From Theorem 4.1, there exists a direction $v \in T_z \mathcal{Z}$ such that $v^\top H_{\mathrm{eff}} v < 0$. Under Assumption 4, the Jacobian $J_{f_\theta}$ is surjective. Moreover, since the number of parameters $|\Theta|$ typically far exceeds the representation dimension $d$, the preimage of $v$ is a high-dimensional affine subspace in $T_\theta \Theta$.

Applying a perturbation $u$ such that $J_{f_\theta} u = v$ to the first term yields $u^\top (J_{f_\theta}^\top H_{\mathrm{eff}} J_{f_\theta}) u = v^\top H_{\mathrm{eff}} v < 0$. Regarding the second term, at a collapsed state or random initialization, the backbone second derivatives $\nabla_\theta^2 f$ are structurally uncorrelated with the residual gradient of the head. In high-dimensional parameter spaces, it is generically possible to choose a direction $u$ in the preimage of $v$ such that the negative curvature of the first term dominates the indefinite contribution of the second term. Consequently, the total parameter Hessian $\nabla_\theta^2 \mathcal{L}$ inherits the negative eigenvalue direction. Since the existence of a negative eigenvalue defines a direction of descent even in the presence of a residual gradient, the configuration is dynamically unstable. $\qquad\square$

## B.8. Proof of Theorem 5.1 (Linear Rank Propagation)

*Remark* B.3 (First-order Gradient Flow Approximation). For the remaining proofs, we analyze the optimization process using the continuous-time gradient flow approximation. All statements regarding representation dynamics are derived from the linearization of the network map $f_\theta$ in a local neighborhood of the parameters. Claims are to be interpreted in the sense of first-order approximations to the training dynamics under the smoothness conditions of Assumption 1.

*Proof.* Let $h(z) = Wz$ where $W \in \mathbb{R}^{k \times d}$. Let $\Sigma_z(t) = \mathbb{E}[z(t)z(t)^\top]$ be the covariance of the backbone representations at training time $t$. We decompose the representation space $\mathcal{Z} = \mathbb{R}^d$ into the row space of the head $\mathcal{S}_\| = \mathrm{Row}(W)$ and its nullspace $\mathcal{S}_\perp = \ker(W)$. Any representation vector $z$ can be written as $z = z_\| + z_\perp$.

The loss function $\mathcal{L}$ depends on $z$ only through the projection $h(z) = Wz_\|$. Consequently, the gradient of the loss with respect to the nullspace component is identically zero: $\nabla_{z_\perp} \mathcal{L} = 0$.

Under Assumption 8, the training dynamics include a weight decay term $\lambda \|\theta\|_2^2$. While this acts on parameters, we analyze its effect on representations via the backbone Jacobian $J_{f_\theta} = \partial z / \partial \theta$. Assuming $J_{f_\theta}$ is nondegenerate (full rank) on the support of the data, the parameter decay $\dot{\theta} = -\lambda\theta$ induces a contraction in representation space. Furthermore, since modern backbones $f_\theta$ (e.g., ResNets and ViTs without bias or with BatchNorm) are often locally homogeneous or bias-free, the contraction $\theta \to 0$ implies $z \to 0$ along unprotected directions.

To a first-order approximation for the nullspace component $z_\perp$:

$$\dot{z}_\perp(t) \approx J_{f_\theta} \dot{\theta} \approx -\alpha z_\perp(t),$$

where $\alpha > 0$ is an effective decay rate determined by the singular values of $J_{f_\theta}$ and $\lambda$. This linear differential equation implies $z_\perp(t) \to 0$ as $t \to \infty$.

Thus, asymptotically, the representation $z$ is confined strictly to the row space $\mathcal{S}_\|$. More formally, the limiting support of the measure induced by $z(t)$ lies in a subspace of dimension at most $\mathrm{rank}(W)$. If the loss forces the output covariance $\Sigma_h$ to have rank $r < k$, then the active row space of $W$ aligns with this $r$-dimensional subspace, and $\mathrm{rank}(\Sigma_z) \to r$. In either case, the rank is strictly upper-bounded by the head dimension: $\mathrm{rank}(\Sigma_z) \le k$. $\qquad\square$

## B.9. Proof of Proposition 5.2 (Metric Singularity)

*Proof.* Let $\xi \in \Xi_{\mathrm{cont}} \subseteq \mathbb{R}^m$ parameterize the continuous augmentations, and let $t_\xi$ be the corresponding augmentation operator. We define the augmentation orbit passing through $z$ as the image of the mapping $\gamma(\xi) = f_\theta(t_\xi(x))$. Since $h$ and $t_\xi$ are assumed $C^1$ in a neighborhood of $z$, the Jacobians $J_h(z)$ and $J_\gamma(0)$ are well-defined.

The augmentation tangent space $\mathcal{V}_{\mathrm{aug}}(z)$ is spanned by the partial derivatives of this mapping at the identity ($\xi = 0$):

$$\mathcal{V}_{\mathrm{aug}}(z) = \mathrm{span} \left\{ \frac{\partial \gamma}{\partial \xi_1}, \dots, \frac{\partial \gamma}{\partial \xi_m} \right\} \bigg|_{\xi=0}.$$

Local invariance implies that the projection head output is constant to first order with respect to small changes in $\xi$. Mathematically, $\nabla_\xi (h \circ \gamma)(\xi)|_{\xi=0} = 0$. Applying the chain rule:

$$J_h(z) \cdot J_\gamma(0) = 0.$$

The columns of $J_\gamma(0)$ are precisely the basis vectors $v \in \mathcal{V}_{\text{aug}}(z)$. Thus, every such tangent vector $v$ lies in the kernel of the head Jacobian $J_h(z)$. The pullback metric is defined as $G(z) = J_h(z)^\top J_h(z)$. For any $v \in \mathcal{V}_{\text{aug}}(z)$:

$$v^\top G(z)v = v^\top J_h(z)^\top J_h(z)v = \|J_h(z)v\|_2^2 = \|0\|_2^2 = 0.$$

Since $G(z)$ is PSD, $v^\top Gv = 0$ implies the metric is singular along these directions. $\qquad\square$

### B.10. Proof of Theorem 5.3 (Information Hierarchy)

*Proof.* We quantify information using the FIM with respect to the augmentation parameters $\xi$. We adopt the standard approximation in nonlinear information geometry that the local FIM is proportional to the Gram matrix of the Jacobian ($J^\top J$) under the assumption of additive, isotropic observation noise. In this setting, rank analysis of the FIM is equivalent to rank analysis of the sensitivity matrix $S_y = \partial y/\partial \xi$.

Let $S_z \in \mathbb{R}^{d \times m}$ be the sensitivity of the backbone $z$ to augmentations. Its column space is exactly $\mathcal{V}_{\text{aug}}(z)$. Assuming the backbone is not already invariant to augmentations (a condition guaranteed by the nondegeneracy of the random initialization and the backbone's high capacity), $\text{rank}(S_z) = \dim(\mathcal{V}_{\text{aug}})$.

Let $S_h \in \mathbb{R}^{k \times m}$ be the sensitivity of the projection $h(z)$. By the chain rule:

$$S_h = J_h(z)S_z.$$

From Proposition 5.2, if the head enforces invariance, then $\ker(J_h(z)) \supseteq \text{Col}(S_z)$. This implies that $J_h(z)$ annihilates the columns of $S_z$. In the ideal case of perfect invariance, $S_h = 0$, and the projection carries rank-0 information about $\xi$.

In the case of approximate invariance, the singular values of $S_h$ corresponding to augmentation directions vanish in the limit of minimizing the loss. Applying the rank-nullity theorem to the restriction of the linear map $J_h|_{\mathcal{V}_{\text{aug}}}$:

$$\text{rank}(S_h) = \text{rank}(S_z) - \dim(\ker(J_h) \cap \mathcal{V}_{\text{aug}}).$$

Since invariance requires $\mathcal{V}_{\text{aug}} \subseteq \ker(J_h)$, the second term is $\dim(\mathcal{V}_{\text{aug}})$. Therefore, the FIM at the projection head is strictly rank-deficient relative to that at the backbone:

$$\text{rank}(\mathcal{I}_{h(z)}) \approx \text{rank}(S_h) \leq \text{rank}(S_z) - \dim(\mathcal{V}_{\text{aug}}).$$

$\qquad\square$

## C. Experimental Setup and Implementation Details

### C.1. Overview of Experiments

Table 2 summarizes the experiments, datasets, backbones, and key metrics which we introduce in this appendix.

*Table 2.* Overview of experiments and configurations.

| Experiment | Objective | Dataset(s) | Backbone(s) | Key metrics |
|---|---|---|---|---|
| Collapse instability | SimSiam-style | CIFAR-10/100 | ResNet-18, ViT-Tiny | variance, condition number, update ratios |
| Guillotine + probes | SimCLR | CIFAR-10/100 | ResNet-18, ViT-Tiny | linear/MLP probe accuracy |
| Curvature + orbits | SimCLR | CIFAR-10/100 | ResNet-18, ViT-Tiny | curvature, orbit metrics |
| Head depth ablation | SimCLR | CIFAR-10 | ResNet-18 | probe acc., curvature, orbit metrics |
| Hessian tracking | SimSiam-style | CIFAR-10 | ResNet-18 | $\lambda_{\min}$, variance, condition number |
| Pretrained analysis | Fixed checkpoints | CIFAR-10 test | ResNet-50, ViT-S/16 | rank, curvature, variance, alignment |

Unless otherwise stated, experiments use CIFAR datasets with standard normalization, batch size 512, and three random seeds. For SimSiam-style collapse experiments we use SGD (lr=0.05, momentum=0.9). For SimCLR pretraining and probes we use Adam (lr=$10^{-3}$, temperature $\tau = 0.1$) with 5 epochs for probe training. When relevant, we report mean and standard deviation over seeds.

## C.2. Metric Definitions

To quantify the geometry of the representation space, the following metrics are reported across experiments:

- **Representation variance:** For a batch of normalized representations $z \in \mathbb{R}^{B \times d}$, we report the mean standard deviation across dimensions: $\mathrm{Var}(z) = \frac{1}{d} \sum_{i=1}^{d} \mathrm{Std}(z_i)$.

- **Condition number:** For the covariance matrix of normalized representations, $\kappa = \lambda_{\max}/\lambda_{\min}$, with $\lambda_{\min}$ clamped to $10^{-7}$ to avoid numerical instability during perfect collapse.

- **Relative update size:** To evaluate timescale separation, we compute $\|\eta \nabla w\|_2/\|w\|_2$ independently for the parameters of the projection head and the backbone.

- **Local curvature ($\kappa$):** For a continuous augmentation trajectory $z_t$ (e.g., sweeping rotation angles) over $T$ steps, we estimate the local manifold curvature using central finite differences:

$$\kappa = \frac{1}{T-2} \sum_{t=2}^{T-1} \|z_{t+1} - 2z_t + z_{t-1}\|_2.$$

- **Mean orbit spread:** The mean squared Euclidean distance of points within a single augmentation orbit from their specific orbit centroid, representing the total variance captured by the nuisance transformation.

- **Intra-orbit distance ($D_{\text{intra}}$):** The average pairwise Euclidean distance between all augmented views generated from a single anchor image $x$, measuring the effective width of the nuisance manifold.

- **Inter-class distance ($D_{\text{inter}}$):** The average Euclidean distance between the centroids of different semantic categories, measuring the global separation of semantic clusters: $D_{\text{inter}} = \mathbb{E}_{c \neq c'}[\|\mu_c - \mu_{c'}\|_2]$.

- **Class/orbit ratio:** A geometric signal-to-noise ratio defined as $D_{\text{inter}}/(D_{\text{intra}} + \epsilon)$. A higher ratio indicates a space where semantic categories are better separated relative to the variance induced by data augmentations.

- **Effective rank (dim):** To determine the effective dimensionality of an augmentation orbit, we compute the entropy $H$ of the squared singular values $\sigma_k^2$ of the centered trajectory matrix:

$$\mathrm{Rank}(z) = \exp\left(-\sum_k p_k \log p_k\right), \quad \text{where } p_k = \frac{\sigma_k^2}{\sum_i \sigma_i^2}.$$

- **Alignment gain ($\Delta_{\text{cos}}$):** The difference in the mean cosine similarity of augmented samples to the unaugmented anchor image, measured between the head output and the backbone output:

$$\Delta_{\text{cos}} = \mathbb{E}_\xi[\cos(h(f(x)), h(f(t_\xi(x))))] - \mathbb{E}_\xi[\cos(f(x), f(t_\xi(x)))].$$

A positive gain indicates the head increases invariance, while a negative gain implies the head actively decorrelates the views (characteristic of redundancy reduction methods).

## C.3. Experiment-Specific Details

**Collapse instability and ablations.** We isolate the attractor dynamics of the positive term using a SimSiam-style objective (random resized crop and horizontal flip). The network is initialized using standard schemes (Kaiming uniform for ResNet-18, truncated normal for ViT-Tiny). The projection head is then explicitly initialized in a pseudo-collapsed state by scaling its linear weights by a factor of $\alpha = 0.1$. This places the network in a pseudo-collapsed state with representation variance $\mathrm{Var}(z) \in [10^{-5}, 10^{-3}]$. For reference, standard initialization yields $\mathrm{Var}(z) \approx 0.05$–$0.15$, indicating our procedure reduces variance by two orders of magnitude. This approximates the basin of attraction of a collapsed fixed point $z^* = c$ (constant representation) while remaining tractable—exact collapse would require solving an intractable constraint satisfaction problem to find $\theta$ such that $f_\theta(x) = c$ for all $x$. The scaling factor $\alpha = 0.1$ was chosen to balance collapse depth (sufficient to observe instability) with numerical stability (avoiding gradient underflow); values of $\alpha \in [0.05, 0.2]$ yield qualitatively similar dynamics. We ablate activation functions (linear, ReLU, GELU, Swish), BatchNorm (on/off), and learning rate scale (base=0.05, large=0.5, small=0.005) while tracking variance, condition number, and update ratios.

**Guillotine effect and probes.** We pretrain with SimCLR augmentations (crop, flip, rotation, color jitter). To quantify information loss, we freeze the network and train probes to predict the specific rotation angle applied to the image. We train both linear and 2-layer MLP probes on the backbone $z$ and head $h(z)$ to isolate linearly separable information from entangled information.

**Curvature and orbit visualization.** To construct smooth augmentation trajectories for local curvature estimation during pretraining, we sweep rotation from $0°$ to $45°$ in $5°$ increments (10 evenly spaced steps) on $\ell_2$-normalized representations. For the high-dimensional orbit visualization and metric calculations, we sample 5 distinct classes, 3 images per class, and sweep through 12 rotations spanning the full $[0°, 360°)$ circle.

**Head depth ablation.** We train SimCLR models with a Swish-activated projection head of varying depths ($L \in \{1, 2, 3\}$) on CIFAR-10 using ResNet-18. We compute linear probe accuracy (evaluating both the backbone and the head), manifold curvature, and orbit compression metrics for each depth architecture.

**Hessian tracking.** We estimate the minimum eigenvalue ($\lambda_{\min}$) of the effective Hessian without materializing the full matrix. We compute exact Hessian-vector products iteratively via double backpropagation, using shifted power iteration (estimating $\lambda_{\max}$ and computing the dominant eigenvector of the shifted operator $H_{\text{shift}}(v) = Hv - \lambda_{\max}v$) with 20 iterations per estimate. Estimates are computed on the first few batches of each epoch.

**Pretrained checkpoint analysis.** We analyze public, pretrained checkpoints where the projection head is available (DINO ViT-S, DINO ResNet-50, VICReg ResNet-50, Barlow Twins ResNet-50) using CIFAR-10 test images resized to $224 \times 224$ with standard ImageNet normalization. To compute unbounded geometric distortions, we sweep four separate continuous orbits on the raw, unnormalized representations: rotation $[0°, 45°]$, hue $[-0.4, 0.4]$, saturation $[0, 2]$, and Gaussian blur $\sigma \in [0.1, 3.0]$, using 12 interpolation steps for each. We report mean statistics over 50 sampled image trajectories and provide standard error intervals for collapse ratios.

## D. Extended Empirical Results

### D.1. Empirical Verification of Timescale Separation

A load-bearing assumption for our stability analysis (Assumption 7) is that the projection head adapts faster than the backbone. To empirically ground this, we logged the relative update magnitudes $\frac{\eta \|\nabla w\|_2}{\|w\|_2}$ for both the head and the backbone during early training (Table 3).

*Table 3.* **Timescale separation ratios.** The ratio of the relative update magnitude of the projection head to the backbone ($\frac{\eta \|\nabla w\|_2}{\|w\|_2}$). Without BatchNorm, the projection head natively adapts orders of magnitude faster than the backbone. Introducing BatchNorm suppresses this natural timescale separation to near-parity across all learning rates.

| Activation | Configuration | Ratio (Head / Backbone) |
|---|---|---|
| Linear | Base LR | 118.5× |
| GELU | Base LR | 56.3× |
| Swish | Base LR | 90.3× |
| | Base LR + BN | 0.9× |
| | Large LR | 66.7× |
| | Large LR + BN | 0.5× |
| | Small LR | 446.6× |
| | Small LR + BN | 2.7× |
| ReLU | Base LR | 2544.1× |
| | Base LR + BN | 0.9× |
| | Large LR | 4234.8× |
| | Large LR + BN | 0.6× |
| | Small LR | 876.2× |
| | Small LR + BN | 1.6× |

When BatchNorm is removed (approximating pure continuous flow), we observed that timescale separation naturally emerges to a massive degree: the projection head updates are orders of magnitude larger relative to its weights than the backbone updates (e.g., $90\times$ for Swish, $56\times$ for GELU, and over $2500\times$ for ReLU). Interestingly, applying BatchNorm artificially suppresses this ratio to near-parity ($< 1\times$ to $3\times$ across all learning rates). This demonstrates that massive

timescale separation is a native property of the gradient flow, whereas normalization heuristics can heavily manipulate and constrain these dynamics.

## D.2. Empirical Verification of Residual Gradients and the Robustness of Swish

To validate Assumption 6, we tracked the residual gradient norms $\|\nabla_h \mathcal{L}\|_2$ near the pseudo-collapsed state. As shown in Figure D.2 (top left), for most heads, the residual gradient norm remains bounded away from zero.

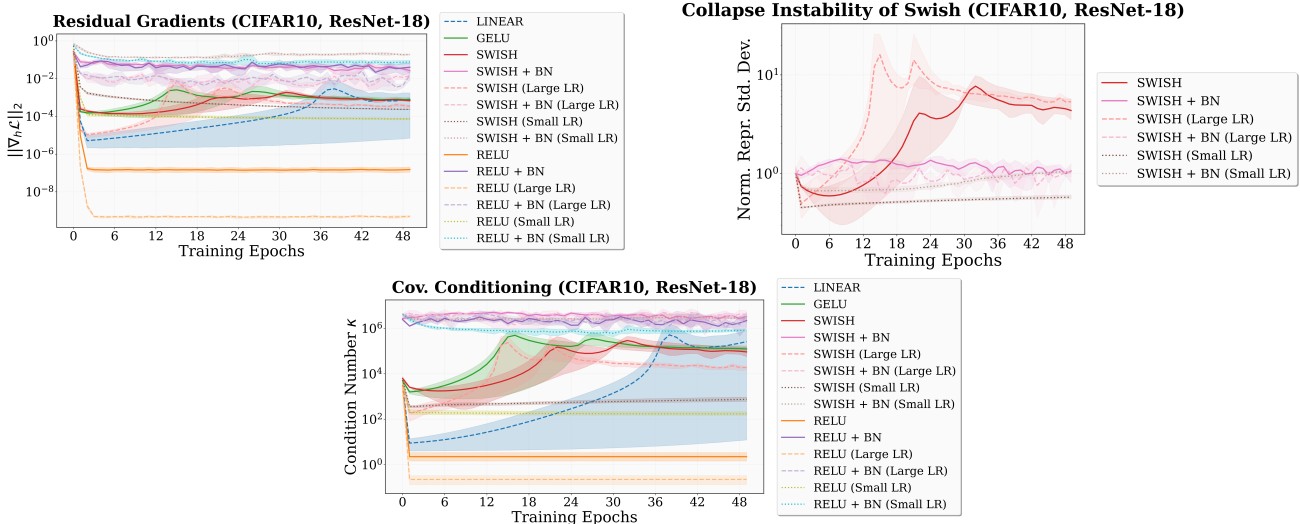

*Figure 6.* **Residual gradients and Swish instability (CIFAR-10, ResNet-18).** Most heads maintain nonvanishing residual gradients (top left), satisfying Assumption 6. Unlike ReLU, Swish can escape collapse across all learning rates and normalization settings (top right). A healthy projection head should have reasonably high condition number (bottom) in order to warp the geometry of the space. This is observed across most tested configurations.

Furthermore, expanding on Section 6, we ran ablations on Swish with other configurations of learning rates and BatchNorm. As shown in Figure 6 (top right), Swish is able to begin escaping with small steps and without BatchNorm, unlike ReLU which had completely flat representation variance. This provides evidence that its geometry natively solves the collapse problem without relying on discretization noise or explicit variance bounds.

## D.3. Robustness to Semantic Density: CIFAR-100

We replicated these tests on CIFAR-100 to ensure the results were not artifacts of low category density. This dataset presents a more challenging optimization landscape with ten times the category density of CIFAR-10, requiring more fine-grained feature preservation in the backbone. Table 4 summarizes the geometric and information-theoretic signatures observed.

*Table 4.* **Metrics for CIFAR-100 (ResNet-18).** Statistics are reported as mean $\pm$ standard deviation. Curvature metrics are calculated over $n = 3$ independent training seeds; probing results are from the final 50-epoch SimCLR representative checkpoint. The consistency of the curvature ratio ($1.84\times$) with CIFAR-10 ($1.93\times$) suggests some universal geometric mechanism.

| Metric Type | Metric | Backbone ($z$) | Head ($h(z)$) | Ratio / Change |
|---|---|---|---|---|
| *Geometric* | Local Curvature | $0.127 \pm 0.002$ | $0.235 \pm 0.001$ | **$1.84\times$ Higher** |
| *Information* | Linear Probing Acc. | 42.01% | 31.19% | -10.82% Abs. Loss |
| | MLP Probing Acc. | 44.53% | 34.16% | -10.37% Abs. Loss |
| | Nonlinearity Gap ($\triangle_{\text{MLP-Lin}}$) | +2.52% | **+2.97%** | **+0.45% Entanglement** |

As shown in Figure 7, the gap between ReLU and nonlinear heads mostly persists on CIFAR-100. Despite the increased task complexity, ReLU heads remain trapped in the collapsed equilibrium (zero representation variance); linear heads seem to recover to the baseline after 20 epochs, but note that the initial decrease in the first few epochs is much steeper than nonlinear heads. In contrast, smooth nonlinearities (GELU, Swish) successfully destabilize the collapsed state, triggering an escape. Interestingly, the escape trajectory for GELU is more aggressive on CIFAR-100 than on CIFAR-10, suggesting that the interaction term in (1) may scale with the complexity of the data distribution.

**Collapse Instability (CIFAR100, ResNet-18)**

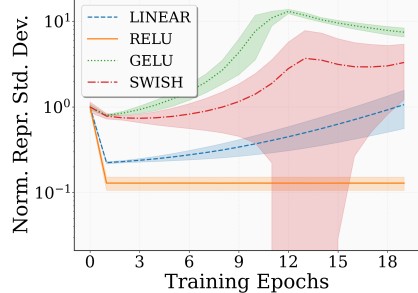

*Figure 7.* **Collapse instability (CIFAR-100).** Evolution of representation variance during training of 3 seeds. Smooth nonlinearities (GELU, Swish) possess the necessary curvature to destabilize the equilibrium and escape the collapsed basin. While the linear head exhibits recovery within these 20 epochs, it only ends up reaching its initial representation variance. Further, the initial decrease in representation variance is steeper for zero-curvature heads than smooth heads.

The information hierarchy observed on CIFAR-10 remains highly significant on CIFAR-100. While the base accuracy for the rotation task is lower on CIFAR-100 due to increased semantic complexity, the guillotine effect is clearly preserved. This confirms that the projection head acts as a consistent information bottleneck across different levels of semantic density, effectively filtering out rotation information that the backbone chooses to retain. The gap between linear and MLP probing accuracy continues to increase in the head relative to the backbone ($+2.97\%$ vs $+2.52\%$), indicating that the head entangles nuisance variables into its higher-order geometry, though less so than CIFAR-10 in our experiment.

Strikingly, the curvature ratio remains remarkably stable across datasets. While CIFAR-10 exhibited a ratio of $1.93\times$, CIFAR-100 yields a nearly identical ratio of $1.84\times$. Despite the drastic change in the number of classes and the base task difficulty, the increase in curvature remains. This provides strong empirical evidence for Proposition 5.2, suggesting that the projection head achieves invariance through a specific, quantifiable topological folding of the representation manifold that is characteristic of the MLP architecture itself.

### D.4. Universality and Stiffness in ViTs

To test the backbone-invariance of our geometric theory, we replicated our experiments using a vision transformer (ViT-Tiny) architecture. This comparison is particularly salient as ViTs lack the convolutional inductive biases and the standard usage of BatchNorm (often cited as a collapse-avoidance mechanism) found in ResNets.

As shown in Table 5, the geometric mechanisms identified in CNNs are not only present in ViTs but are significantly amplified. While qualitatively consistent with our ResNet findings, the ViT results are quantitatively distinct in two key ways. The first is the more aggressive orbit compression. While the ResNet-18 head reduced orbit spread by $21.85\times$, the ViT head collapses it by a remarkable $4015.33\times$. We attribute this to the ViT backbone which, without the convolutional prior to pre-filter spatial nuisances, has augmented views which remain highly dispersed. Consequently, the projection head must induce a significantly more singular metric to achieve the invariance required by the SSL objective. This is reflected in the higher curvature ratio ($2.59\times$ in ViTs compared to $1.93\times$ in CNNs), indicating the head must warp the manifold more aggressively.

Secondly, despite this extreme destruction of variance, the head remains a selective filter. As shown in Table 5, the intra-orbit noise ($D_{\text{intra}}$) is crushed by $64.13\times$, whereas the inter-class signal ($D_{\text{inter}}$) is reduced by only $55.95\times$. This asymmetry results in a $1.14\times$ improvement in the class/orbit ratio. That is, metric singularity specifically targets augmentation directions, preserving semantic separation even when the majority of the representation variance is collapsed.

Notably, while Theorem 4.1 proves that the collapsed state remains a strict unstable region for ViTs, we observed that the transformer backbone exhibited significant optimization stiffness compared to ResNets. In our 20-epoch window, all projection heads remain trapped near the collapsed equilibrium (Figure 8). This suggests that while the geometric instability exists (as theoretically proven), the lack of BatchNorm's landscape-smoothing properties (Santurkar et al., 2018) and the sharp nature of the ViT loss landscape (Chen et al., 2021; 2022) make these instabilities harder to escape via first-order methods.

These results as a whole demonstrate why ViTs are promising for geometric SSL. Unlike static convolutional hierarchies, the

*Table 5.* **Metrics for CIFAR-10 (ViT-Tiny).** The results demonstrate that the projection head acts as a universal geometric filter across backbones, independent of inductive biases.

| Metric Type | Metric | Backbone ($z$) | Head ($h(z)$) | Ratio / Change |
|---|---|---|---|---|
| *Geometric* | Local Curvature | $0.123 \pm 0.0002$ | $0.317 \pm 0.002$ | **2.59$\times$ Higher** |
| | Mean Orbit Spread ($\times 10^{-2}$) | $1.09 \pm 0.37$ | $0.0003 \pm 0.0001$ | **4015.33$\times$ Collapse** |
| | Intra-orbit Distance ($D_{\text{intra}}$) | $0.148 \pm 0.024$ | $0.002 \pm 0.0004$ | $64.13\times$ Reduction |
| | Inter-class Distance ($D_{\text{inter}}$) | $0.313 \pm 0.039$ | $0.006 \pm 0.001$ | $55.95\times$ Reduction |
| | Class/Orbit Ratio ($D_{\text{inter}}/D_{\text{intra}}$) | $2.12 \pm 0.44\times$ | $2.42 \pm 0.58\times$ | $1.14\times$ Separation |
| *Information* | Linear Probing Acc. | 40.75% | 34.92% | -5.83% Abs. Loss |
| | MLP Probing Acc. | 44.22% | 40.87% | -3.35% Abs. Loss |
| | **Nonlinearity Gap ($\triangle_{\text{MLP-Lin}}$)** | **+3.47%** | **+5.83%** | **+2.48% Entanglement** |

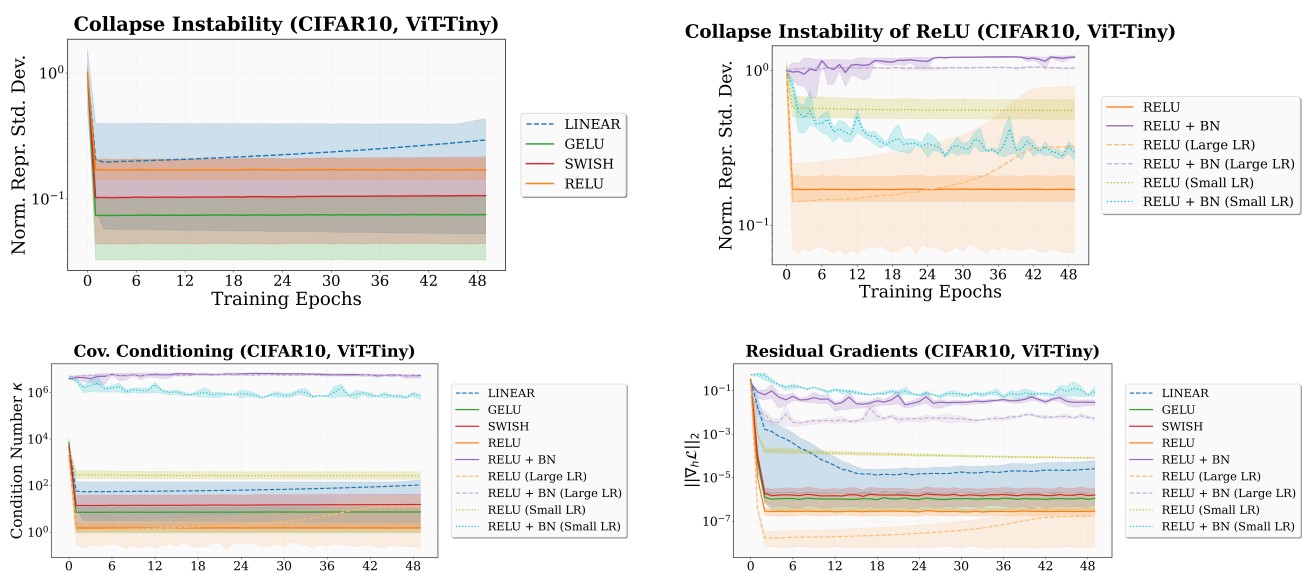

*Figure 8.* **Optimization geometry (CIFAR-10, ViT-Tiny).** Unlike ResNets, the ViT backbone is exceptionally stiff. For all activations, representation variance (top left) remains trapped near zero. It is observed that adding BatchNorm for ReLU (top right) allows for some attempt at escape. The condition numbers (bottom left) remain mostly flat. Finally, the residual gradients are still bounded away from zero for most activations, but generally smaller than in CNNs. Overall, while smooth heads are theoretically necessary for stability, they are not alone sufficient to overcome ViT landscape sharpness without extended epochs or backbone regularization.

ViT backbone generates a data-dependent Jacobian via self-attention. The results show that even when the backbone geometry is dynamic, the projection head consistently induces a singular pullback metric to satisfy the invariance objective. This emphasizes the necessity of smooth activations like GELU (Hendrycks & Gimpel, 2023): they provide the intrinsic curvature required to theoretically destabilize the collapsed state, even when the structural stiffness of the ViT backbone (Súkeník et al., 2025) severely dampens the observable escape dynamics.

Our theory may offer a geometric explanation for the industry-wide shift from piecewise-linear activations (e.g., ReLU) to smooth nonlinearities (e.g., GELU, Swish) in SSL more generally. While early contrastive methods like SimCLR successfully used ReLU, they relied on large batch sizes and specific normalization layers to maintain variance. In contrast, modern frameworks (e.g., DINO, MAE) and transformer architectures favor GELU. Our results suggest that this shift provides an intrinsic geometric benefit which makes them more stable, reducing reliance on careful batch-size scaling.

### D.5. Projection Head Depth Ablation

As discussed in Section 3.2 and Proposition 3.4, the projection head must possess sufficient geometric capacity to successfully isolate the metric singularity required for invariance. To empirically validate this, we ablated the depth of a Swish-activated projection head on CIFAR-10 using a ResNet-18 backbone, evaluating depth configurations from a purely linear projection ($L = 1$) to a (deeper) nonlinear MLP ($L = 3$).

Table 6 exposes the mechanics of the guillotine effect. A shallow linear head forces a rank bottleneck that bleeds upstream;

*Table 6.* **Projection head depth ablation (CIFAR-10, ResNet-18).** As head depth increases, both local manifold curvature and orbit compression intensify. Deeper heads possess the geometric capacity to violently warp the space to absorb the metric singularity, sacrificing their own linear separability to protect the upstream backbone.

| Head Architecture | Head Acc. | Backbone Acc. | Curvature Ratio | Orbit Comp. |
|---|---|---|---|---|
| Linear ($L = 1$) | 64.2% | 68.1% | 1.08$\times$ | 49.1$\times$ |
| 2-Layer MLP ($L = 2$) | 56.5% | 73.4% | 1.93$\times$ | 57.2$\times$ |
| 3-Layer MLP ($L = 3$) | 51.1% | **75.2%** | **2.54$\times$** | **78.5$\times$** |

*Table 7.* **Geometric analysis of pretrained foundation models.** Comparison of backbone ($f$) vs. projection head ($h$) geometry. Rank measures the effective dimensionality of the orbit. Variance measures the total spread (scaled by $10^{-2}$). Collapse quantifies the compression of the augmentation manifold (in terms of variance of representations between the head and backbone). Curvature measures the warping of the manifold. Gain ($\Delta_{\cos}$) measures the improvement in cosine invariance induced by the head.

| Model | Orbit | Dimensionality (Rank) Backbone | Head | Variance ($\times 10^{-2}$) Backbone | Head | Collapse Ratio (Mean $\pm$ CI) | Curvature Back. | Head | Ratio | Alignment Gain ($\Delta_{\cos}$) |
|---|---|---|---|---|---|---|---|---|---|---|
| **DINO ViT-S** | Rotation | 3.65 | 2.72 | 512.2 | 0.42 | **1284.4 $\pm$ 62.0** | 39.1 | 14.0 | **0.36$\times$** | +0.34 |
| | Hue | 6.50 | 3.07 | 224.7 | 0.13 | **2498.6 $\pm$ 321.3** | 43.9 | 12.8 | **0.29$\times$** | +0.23 |
| | Saturation | 2.45 | 2.36 | 200.2 | 0.09 | **2898.1 $\pm$ 435.7** | 11.3 | 3.5 | **0.31$\times$** | +0.32 |
| | Blur | 2.01 | 1.73 | 90.5 | 0.05 | **3727.7 $\pm$ 857.9** | 4.6 | 1.4 | **0.30$\times$** | +0.04 |
| **DINO ResNet50** | Rotation | 4.35 | 2.39 | 0.35 | 0.00 | **176.2 $\pm$ 26.1** | 2.76 | 1.26 | **0.45$\times$** | +0.41 |
| | Hue | 6.72 | 3.70 | 0.13 | 0.00 | **223.1 $\pm$ 31.6** | 2.71 | 1.06 | **0.41$\times$** | +0.25 |
| | Saturation | 2.60 | 2.20 | 0.08 | 0.00 | **116.9 $\pm$ 22.3** | 0.68 | 0.40 | **0.61$\times$** | +0.31 |
| | Blur | 1.86 | 1.66 | 0.17 | 0.00 | **227.2 $\pm$ 29.3** | 0.60 | 0.24 | **0.41$\times$** | +0.11 |
| **VICReg ResNet50** | Rotation | 3.82 | 3.32 | 4.71 | 4.60 | 1.4 $\pm$ 0.2 | 8.51 | 18.28 | **2.15$\times$** | -0.37 |
| | Hue | 6.71 | 4.50 | 2.22 | 1.66 | 2.1 $\pm$ 0.5 | 11.73 | 19.37 | **1.66$\times$** | -0.40 |
| | Saturation | 2.56 | 2.18 | 1.34 | 1.73 | 1.1 $\pm$ 0.2 | 2.68 | 6.13 | **2.31$\times$** | -0.37 |
| | Blur | 1.79 | 1.75 | 0.84 | 1.54 | 0.9 $\pm$ 0.2 | 1.10 | 3.19 | **2.88$\times$** | -0.04 |
| **Barlow Twins** | Rotation | 3.82 | 3.48 | 0.35 | 0.18 | 2.6 $\pm$ 0.3 | 2.20 | 4.03 | **1.83$\times$** | -0.36 |
| | Hue | 7.06 | 4.42 | 0.14 | 0.07 | 2.5 $\pm$ 0.3 | 2.98 | 4.39 | **1.49$\times$** | -0.39 |
| | Saturation | 2.75 | 2.63 | 0.09 | 0.07 | 2.5 $\pm$ 0.7 | 0.79 | 1.49 | **1.90$\times$** | -0.33 |
| | Blur | 1.82 | 1.72 | 0.06 | 0.07 | 1.4 $\pm$ 0.3 | 0.31 | 0.70 | **2.24$\times$** | -0.05 |

it fails to adequately warp the space (curvature ratio of $1.08\times$), resulting in weak orbit compression ($49.1\times$) and relatively worse downstream backbone accuracy ($68.1\%$). However, as depth increases to 3 layers, the head increases its capacity to warp the local manifold (curvature ratio jumps to $2.54\times$). By acting as a highly capable Riemannian preconditioner, the deeper head better isolates the singularity, causing extreme orbit compression ($78.5\times$) in its own latent space, which drops its linear probing accuracy to $51.1\%$, while insulating the backbone and increasing downstream accuracy to $75.2\%$.

While it is recommended to have a projection head architecture which has sufficient expressive capacity as an approximator, practitioners should choose an architecture which equilibrates to the backbone fast to ensure timescale separation. The current trend of 2- or 3-layer MLPs seems to meet that criteria.

### D.6. An Analysis of Pretrained Foundation Models

Finally, to show the persistence of these effects at scale we consider a range of pretrained models where the projection head is available (DINO (Caron et al., 2021), VICReg (Bardes et al., 2022), Barlow Twins (Zbontar et al., 2021)). Table 7 reveals a geometric dichotomy between clustering-based and redundancy-reduction methods.

The results for DINO provide the strongest empirical validation of the metric singularity hypothesis. Across all augmentation orbits, the projection head induces extreme geometric compression. For the ViT-S backbone, the rotation orbit spread is reduced by a factor of $1284\times$, and hue variance is crushed by nearly $2500\times$. This massive collapse is accompanied by a reduction in effective rank (from $3.65$ to $2.72$ for rotation), indicating that the head is actively flattening the high-dimensional augmentation manifold into a lower-dimensional subspace. This geometric destruction yields a strictly positive alignment gain ($\Delta_{\cos} > +0.2$ for most geometric orbits); here, the head's primary role is to enforce invariance by mapping distinct augmented views to a single point. Notably, the curvature ratio is consistently below unity, implying that the head not only shrinks the augmentation orbits but actively straightens them. By smoothing out the high-frequency curvature of the backbone trajectory, the head simplifies the complex geometric path into a linear interpolation towards the cluster centroid.

In contrast, redundancy reduction methods exhibit a completely different geometric signature. The collapse ratios are near unity ($1$–$2\times$), indicating that the projection head preserves the total variance of the augmentation orbits. Most strikingly,

the alignment gain is consistently negative ($\Delta_{\cos} \approx -0.35$). This implies that the projection head makes the augmented views less similar to each other than they were in the backbone. The loss functions for VICReg and Barlow Twins explicitly penalize covariance correlations to prevent collapse. To satisfy this full-rank constraint without destroying the semantic structure of the backbone, the projection head acts as a dimensional expander of sorts: it absorbs the decorrelation stress by actively separating views (negative gain) and increasing manifold complexity (e.g., curvature increases from $8.51$ to $18.28$ for VICReg rotation), thereby allowing the backbone to remain semantically compressed and invariant.

Despite these opposing mechanisms, the projection head serves a universal purpose: it decouples the semantic backbone from the rigid geometric constraints of the training objective. Whether the objective demands extreme invariance (DINO) or extreme variance (VICReg), the MLP head provides the topological flexibility to satisfy these constraints—via singularity or expansion—shielding the backbone's linear separability. Future work may investigate the complementary question of how the backbone geometry itself adapts when this decoupling mechanism is removed or altered.

