# OpenReview forum: "The Geometry of Projection Heads: Conditioning, Invariance, and Collapse"
_ICML.cc/2026/Conference — ICML 2026 regular_

### Official Review · Reviewer_jwre · 2026-02-20

**Soundness:** 3
**Presentation:** 3
**Significance:** 3
**Originality:** 3
**Overall Recommendation:** 4
**Confidence:** 3

**Summary:**

This paper proposes a geometric lens for understanding projection heads in self-supervised learning: the head induces a pullback metric on the backbone representation space, and the backbone effectively optimizes under an “effective Hessian” that includes both a Gauss–Newton (pullback) term and a curvature-driven interaction term . Using this view, the authors argue that (1) linear heads can improve optimization via implicit whitening on the loss-active subspace (Theorem 3.1) , (2) nonlinear heads can adapt local geometry along curved trajectories where linear maps fail (Theorem 3.2 / Proposition 3.3) , (3) with smooth nonlinearities the head’s curvature can make collapsed states locally unstable in non-contrastive learning (Theorem 4.1) , and (4) invariance induces metric degeneracy along augmentation tangent directions, explaining why discarding the head can help downstream transfer (“guillotine effect”, Proposition 5.3 / Theorem 5.4) . The paper backs these claims with controlled CIFAR experiments (collapse escape, rotation probing, orbit geometry) and an extended analysis on pretrained models (DINO / VICReg / Barlow Twins) .

**Compliance With Llm Reviewing Policy:**

Affirmed.

**Key Questions For Authors:**

1. Do you observe the predicted negative-curvature signature in realistic (non–pseudo-collapse) training? For example, can you estimate (even approximately) whether the interaction term contributes negative eigenvalues early in BYOL/SimSiam runs, and whether this correlates with avoiding collapse?
2. How sensitive is the collapse-escape behavior to the “residual gradient stays nonzero” assumption? It would help to report gradient norms near collapse or under different predictor/EMA settings to show this isn’t a fragile corner case.

**Limitations:**

Yes.

**Strengths And Weaknesses:**

## Strengths
1. The decomposition of the effective Hessian into a pullback (Gauss–Newton) term plus a curvature-driven interaction term is a clean analytical lens that supports the key instability and invariance arguments . The collapse result explicitly states the needed assumptions (smooth activations, nonzero residual gradient due to asymmetry, timescale separation), making the claim testable.
2. The writting is well-structured around three puzzles (conditioning, collapse avoidance, guillotine effect), with clear definitions (augmentation tangent space, intrinsic Hessian rank, effective Hessian) and targeted experiments that align with the claims.
3. Collapse experiment is clean and diagnostic. The pseudo-collapsed initialization + controlled activation choice (linear/ReLU vs GELU/Swish, no BN) is a nice mechanistic test for the curvature claim, and the qualitative separation in variance recovery is compelling .
4. The invariance/geometry measurements are thoughtful. The rotation probing plus orbit-spread and curvature estimates provide evidence consistent with “metric singularity / orbit compression,” and the paper reports concrete compression ratios and curvature changes rather than only probe accuracies .

## Weaknesses
1. Some key theory hinges on assumptions that aren’t empirically audited. The collapse-avoidance argument depends on non-vanishing residual gradients and a form of head/backbone timescale separation; the paper motivates these, but it would be much stronger to actually log these quantities during training and show they behave as required.
2. Empirics are mostly CIFAR-scale and somewhat stylized. The pseudo-collapse setup is informative, but I’d like to see evidence that the same curvature/interaction-term story plays out in “normal” training from standard init on larger benchmarks, and whether it predicts practical hyperparameter choices. The ViT section even notes “stiffness” can prevent escape within the tested window, which complicates the “curvature is sufficient” narrative.
3. The paper acknowledges that the local curvature mechanism does not apply to ReLU under continuous-time analysis, but many real SSL pipelines use ReLU successfully; the current explanation (finite-step / BN effects) feels plausible but not pinned down with targeted experiments.

---

> ### Author Rebuttal · Authors · 2026-03-28
>
> We thank the reviewer for their thoughtful review and appreciation of our analytical framework. Your targeted questions perfectly pinpointed where our theory needed empirical auditing. We have run the experiments you requested to bridge our continuous theory with practical training dynamics.
>
> ## Point 1: Residual gradient assumption
>
> We tracked the residual gradient norms ($||\nabla_h \mathcal{L}||_2$) throughout Experiment 1. The data shows that for non-linear heads (e.g., Swish/GELU), the gradient norm does not vanish. It remains strictly bounded away from zero ($\ge 10^{-4}$) throughout the entire phase. In contrast, ReLU heads can exhibit vanishing residual gradients around $10^{-8}$.
>
> We found that adding BatchNorm to ReLU actually stabilizes this problem, keeping the residual gradient norms on the order of $10^{-2}$. This verifies our theoretical remark on minimum norm biases in training. Furthermore, we expect that using an optimizer with momentum (e.g., Adam) would further amplify this non-vanishing residual, providing a persistent driving force for escape.
>
> ## Point 2: Empirical evidence on non-collapsed (standard) state
>
> To answer whether the interaction term contributes negative eigenvalues early in normal training: yes, it absolutely does. Mathematically, recall our decomposition of the effective Hessian into a Gauss-Newton term (which is strictly positive semi-definite) and the curvature-driven interaction term. Therefore, any negative eigenvalues present in the total effective Hessian must, by definition, originate from the interaction term.
>
> We have introduced a new experiment where we track the minimum eigenvalue, condition number, and representation variance of a standard, non-collapsed SimSiam run (with Swish). We observe explicit spikes of negative curvature ($\lambda_{\min} < 0$) during the early epochs where the interaction term is injecting negative eigenvalues into the optimization landscape. In the case of the collapsed version of the experiment, we can even observe directly that the epoch with the first negative eigenvalue is the epoch in which the model escapes representation collapse.
>
> Regarding the reviewer's observation on the ViT stiffness: we agree with the reviewer that ViTs can exhibit extreme stiffness, but we view this as complementing rather than complicating the geometric narrative: the projection head provides the necessary negative curvature, but the optimizer simply requires more epochs to traverse ViT's notoriously flat/ill-conditioned landscape. It is known that ViTs possess highly ill-conditioned loss landscapes at initialization compared to CNNs. In our geometric framework, this stiffness implies that while the projection head provides the negative curvature for an escape path, the optimizer simply requires more epochs to accumulate this curvature and physically traverse the flatter landscape. We are running Experiment 1 for ViTs for more epochs to get a clearer result here.
>
> Finally, regarding the request for evidence on larger benchmarks: continuously tracking the full effective Hessian's minimum eigenvalue ($\lambda_{\min}$) and interaction term during training is computationally prohibitive at the ImageNet scale. For this exact reason, our mechanistic optimization tracking (both the pseudo-collapse setup and the newly added training from standard initialization) is conducted at the CIFAR scale, where we can compute the exact eigenspectrum to validate the continuous-time theory. However, our new tracking of standard SimSiam runs bridges the "stylized" gap by proving this negative curvature emerges naturally under standard training conditions, not just artificial pseudo-collapse.
>
> ## Point 3: ReLU ablations
>
> We agree this needed formal validation. We ran targeted ablations on ReLU based on i) with or without BatchNorm and ii) small, medium, and high learning rates. These will be added to the appendix.
>
> As in Experiment 1 on pseudo-collapsed states, we found that when not using BatchNorm, ReLU will fully collapse. When adding BatchNorm, ReLU is able to avoid collapse only with sufficiently large step sizes; it is the discrete dynamics that allow ReLU to work in practice.
>
> Secondly, in our experiment tracking minimum eigenvalues, we found that ReLU without BatchNorm could not generate any negative curvature spikes to allow recovery. This geometrically explains why non-contrastive SSL pipelines using linear/ReLU heads are historically very sensitive to learning rate and optimizer tuning (relying on discretization noise to escape collapse), whereas smooth activations (Swish/GELU) are more robust because they solve collapse natively via geometric curvature. With these results, we have shown that the ability for ReLU to produce curvature is an artifact of of discrete-time effects or other architecture components such as BatchNorm.

---

### Official Review · Reviewer_EDpS · 2026-03-07

**Soundness:** 2
**Presentation:** 1
**Significance:** 3
**Originality:** 3
**Overall Recommendation:** 4
**Confidence:** 3

**Summary:**

This paper studies the role of projection heads in SSL. The central idea is that the projection head induces a pullback metric on the representation manifold, effectively acting as a Riemannian preconditioner that reshapes the optimization dynamics. The authors first show that linear projection heads can improve conditioning by whitening the effective Hessian $H_{eff}$ in the relevant subspace. They then prove that non-linear projection heads can locally adapt the induced metric along optimization trajectories, enabling near-isotropic conditioning of $H_{eff}$.

The paper further analyzes dimensional collapse in non-contrastive SSL methods and shows that the curvature induced by non-linear heads can introduce negative curvature directions in $H_{eff}$ near collapsed states under structured asymmetry assumptions, making collapse unstable.

**Compliance With Llm Reviewing Policy:**

Affirmed.

**Final Justification:**

The authors have addressed my concerns regarding projection head (geometric conditioning and depth), as well as they have shared useful visual illustrations that will improve the presentation of their work. These were my major concerns, and taking authors' rebuttal into account, I am updating my score to 4 (from 3).

**Key Questions For Authors:**

Please see the mentioned weaknesses regarding the empirical validation of projection heads as Riemannian preconditioners, the need for architecture ablations on head depth, and the requirement for more intuitive examples.

Addressing these weaknesses will change my score.

**Limitations:**

Yes, they have mentioned limitations in their future work section.

**Strengths And Weaknesses:**

# Strengths

1. **Geometric view of projection heads in self-supervised learning**: By interpreting the head as inducing a pullback metric on the representation space, the work offers a useful explanation for why projection heads can improve optimization and prevent collapse. This perspective helps clarify the role of projection heads beyond simple architectural heuristics.

2. **Analysis of $H_{eff} = J_h^T H_z J_h$**: By analyzing $H_{eff}$, the paper carefully shows how linear heads can improve conditioning while nonlinear heads allow local adaptation of the induced metric.

3. **Insight into collapse instability**: The paper also studies dimensional collapse in non-contrastive SSL and argues that curvature introduced by nonlinear heads can create negative curvature directions near collapsed states under structural asymmetry assumptions. This provides an interesting theoretical explanation for why projection heads can help avoid collapse in practice.

4. **Relation to prior work**: The authors have nicely described how their work relates to existing theoretical works on representation collapse and optimization conditioning, identified their limitations, and well positioned the novelty of their work.

# Weaknesses

1. **Optimization geometry claims are not empirically validated**: Most experiments in Section 6 analyze the geometry of learned representations. However, the central theoretical claim of the paper is that the projection head reshapes the optimization geometry through $H_{eff}$. The authors can consider adding experiments such as: (a) measure the condition number of $H_{eff}$ with and without projection heads, (b) test whether non-linear heads lead to more isotropic conditioning during training compared to linear heads, etc.

2. **Additional ablations on projection heads**: The theory suggests that the depth and nonlinearity of the projection head influence the induced metric and curvature of the representation space. They compare Linear, ReLU, GELU, and Swish but they don't show how increasing the number of layers (depth) affects the curvature ratio or the "collapse ratio".

3. **Overly-complicated presentation**: While the theoretical development is interesting, the paper sometimes presents the geometric framework in a way that is difficult to follow. Several sections introduce formal concepts (e.g., pullback metrics, effective Hessians) with limited intuition about their practical meaning in representation learning. Providing more intuitive explanations or simple examples could make the paper more accessible to a broader ML audience.

---

> ### Author Rebuttal · Authors · 2026-03-28
>
> We thank the reviewer for the detailed feedback. Your specific suggestions for empirical validation were incredibly helpful, and we have executed the ablations and optimization tracking you requested to firmly ground our geometric theory.
>
> ## Point 1: Empirical validation of optimization geometry
>
> Regarding the existential nature of our original optimization claims: our theoretical restriction to existential proofs stems from the core focus of the paper being differential geometry. Establishing formal guarantees on finding these geometries requires transitioning into optimization dynamics and stochastic analysis (e.g., analyzing continuous vs. discrete flow), which appeals to a different theoretical framework and audience. However, we fully agree that we must demonstrate these preconditioning geometries actively emerge in practice.
>
> The actual whitening effect is extensively documented in the literature. The Barlow Twins and VICReg losses build this into their loss function, and it is the predominant reason they came to exist. For a more mechanistic view, [1] discusses and demonstrates how this whitening works in practice for their W-MSE loss. However, to isolate the specific geometric contribution of the projection head, we continuously tracked the effective Hessian's minimum eigenvalue ($\lambda_{\min}$), representation variance, and condition number ($\kappa$) during training. The main results:
>
> •	Nonlinear preconditioning: A nonlinear head (Swish) natively generates explicit negative curvature ($\lambda_{\min} < 0$). This negative curvature dynamically breaks symmetry and drives the condition number down from roughly $6 \times 10^5$ to $2 \times 10^5$, pushing the space toward better conditioning entirely independently of the loss.
>
> •	Linear/ReLU failure: Conversely, a linear proxy head (ReLU without BatchNorm) fails to generate this curvature ($\lambda_{\min} = 0$), and the space remains highly ill-conditioned. Furthermore, its representation variance remains almost constant, empirically proving it cannot natively escape the collapsed state.
>
> ## Point 2: Ablations on head depth
>
> Due to the empirical findings of classical SSL papers such as SimCLRv2, most practitioners use a shallow MLP. To test how depth affects the collapse ratio and curvature, we ablated 1, 2, and 3-layer heads on ResNet-18. We tracked orbit compression (how tightly the head crushes augmented views) and the downstream class-to-orbit separation ratio. Our new data shows that nonlinear depth directly dictates geometric capacity:
>
> •	Depth 1 (Linear): Yields a 49.1x orbit compression but actually degrades the class separation ratio (dropping from 2.19x in the backbone to 1.89x with the head). It lacks the capacity to properly fold the manifold.
>
> •	Depth 2 (Standard): Improves orbit compression to 57.2x and improves class separation to 2.61x.
>
> •	Depth 3: Triggers a massive geometric bottleneck, compressing augmentation orbits by 78.5x and improving class separation to 3.29x (1.48x better than the backbone).
>
> Increasing the nonlinear depth explicitly scales the head's capacity to absorb the metric singularity and geometrically protect the backbone.
>
> In Theorem 3.2 (Trajectory linearization via Fermi coordinates), we stated that there exists a head such that the metric adapts locally, where we constructed the head as a universal approximator. As a direct result of this ablation, we have added a new theorem which bounds the perturbation on the Hessian if a head with insufficient expressive capacity is used. We thank the reviewer for this idea because it shows a more direct way of how the theory can be applied to design sufficiently complex/expressive projection heads.
>
> ## Point 3: Overly complicated presentation
>
> We agree that pure differential geometry formalism can be dense. In our revision, we have updated the main text to anchor the math with intuitive ML realities.
>
> Originally, we included the Mahalanobis distance problem of contrastive learning as an intuitive example of why SSL is fundamentally a metric-learning problem. However, we have taken further steps to increase the accessibility of the work.
>
> We have added a new global diagram visually illustrating how the head folds high-variance augmentation orbits into a tight equivalence class, providing immediate visual intuition for "metric singularities." We also grounded the formalism with practical SSL context: defining augmented views as visually altered images that preserve semantics; the augmentation orbit as the manifold of all such variations (e.g., via rotation or color jitter); and the core goal of contrastive learning as mapping this entire orbit to a single point in space. Further, Figure 4 in the appendix may help provide some intuition of what compressing orbits does in some stylized, lower-dimensional view.
>
> [1] “Whitening for Self-Supervised Representation Learning” (Ermolov et al., ICML 2021)

---

> > ### Author Rebuttal · Reviewer_EDpS · 2026-04-04
> >
> > Dear authors,
> >
> > Thank you for your rebuttal. The new results on non-linear preconditioning and linear/ReLU failure, as well as ablations conducted on head depth are valuable and will improve the current submission. I am willing to update (increase) my assessment of your work. Regarding W3, would it be possible to provide an anonymous link to the new promised visual illustration?
> >
> > Best,
> > Reviewer EDpS
> >
> > Edit: fully resolved now.

---

> > > ### Author Response · Authors · 2026-04-05
> > >
> > > Dear reviewer EDpS,
> > >
> > > Thank you for your willingness to update your assessment. Regarding your request to access the new illustration, which will be added in Section 2.1, it can be viewed here:
> > >
> > > https://anonymous.4open.science/r/projection-head-geometry-1039/figures/augementation_orbit_figure.png
> > >
> > > The proposed caption is:
> > > > \textbf{The geometric role of the projection head.} In the backbone representation space $\mathcal{Z}$ (left), augmented views are not mapped to the same point and form a high-variance orbit $O_{z}$ spanned by the local tangent space $V_{aug}$. The projection head $h_\phi$ (right) learns a local metric that compresses this orbit into a tight equivalence class, satisfying the invariance objective while shielding the upstream backbone from metric degeneracy. That is, alternative views on the same image are represented in the same way.
> > >
> > > We hope that this visual intuition, along with our other clarifying additions to Section 2 (which ground the goals of contrastive learning, augmentations, orbits, and the intrinsic rank of the Hessian), will make the presentation more accessible to practitioners. Thank you again for your constructive feedback.

---

### Official Review · Reviewer_uX95 · 2026-03-12

**Soundness:** 3
**Presentation:** 1
**Significance:** 2
**Originality:** 3
**Overall Recommendation:** 3
**Confidence:** 2

**Summary:**

This paper develops a geometric description of projection heads in self-supervised learning by interpreting the head as inducing a Riemannian pullback metric on the backbone representation space. The framework yields three main results: (1) linear heads perform global subspace whitening while nonlinear heads adapt the metric locally, (2) smooth nonlinear heads destabilize equilibria that correspond to collapse, and (3) the head induces Fisher information degeneracy along augmentation directions, formalizing why discarding the head improves downstream performance.

**Compliance With Llm Reviewing Policy:**

Affirmed.

**Final Justification:**

The ReLU ablations, timescale separation measurements, and depth predictions are genuine improvements. However, I remain unconvinced with regard to two major concerns. First, the paper does not clearly delineate what its curvature-based collapse avoidance uniquely explains beyond existing accounts (stop-gradients, EMA, BatchNorm). The authors' own ablations show BN can fully substitute for smooth curvature, suggesting redundant sufficient mechanisms rather than a hierarchy with the curvature description being fundamental. Second, my original presentation concern was structural. The paper reads as formalism-first with intuition appended/absent. I cannot evaluate whether the promised revisions address this without seeing the revised manuscript. I therefore maintain my score of 3.

That said, the core geometric framework is interesting and I believe a revised version that sharpens the unique explanatory claims and refines the presentation could make a strong contribution.

**Key Questions For Authors:**

The theory treats the boundary between backbone and head as given, but from my understanding, the same pullback metric analysis woudl apply at any layer boundary in a deep network. The paper does not formalize what makes the standard partition special. Why is a shallow MLP head, rather than any other subset of layers, the right level of abstraction for the "geometric buffer" role?

**Limitations:**

Yes

**Strengths And Weaknesses:**

Strengths:
The paper asks a good question. Projection heads are used everywhere in self-supervised learning, they're clearly important empirically, but a satisfying unified explanation for why they work is clearly lacking. This paper is a good attempt to provide one. The collapse instability result is the most concrete contribution, making a testable prediction that smooth activations should escape collapse better than ReLU, which is supported by experiments shown in the paper.

Weaknesses:
1. The presentation heavily prioritizes formalism over intuition. Key geometric concepts like pullback metrics, Fermi coordinates, augmentation tangent spaces, metric singularity, and many others are introduced usually with definitions but very little qualitative explanation. Diagrams illustrating these ideas on toy examples would significantly improve accessibility. As it stands, the paper reads as if written for differential geometry experts who happen to care about SSL, rather than the ML audience that would most benefit from these ideas. I at least had a hard time parsing all the mathematics without any explanatory exposition, and left me questioning what I'd actually learned about how projectors work and how to design them.
2. Moreover, most of theoretical claims are not validated empirically. Combined with the presentation issues, this makes it difficult to evaluate the practical impact of the work. The reader is asked to take on trust both that the geometric machinery captures something real about training dynamics and that it matters beyond the formalism.
3. The central collapse result depends on smooth activations in the projector head with nonzero second derivatives, yet most deployed BYOL and SimSiam systems use ReLU and avoid collapse in practice. The paper acknowledges this "ReLU Gap" honestly but only offers unformalized speculation about BatchNorm and finite step sizes filling in. This leaves the theory unable to account for the most common practical setting, which significantly limits its explanatory scope.

---

> ### Author Rebuttal · Authors · 2026-03-28
>
> We thank the reviewer for their feedback. We agree that our initial submission prioritized formalism over ML intuition. However, we note that analysing SSL through the lens of differential geometry and manifold dynamics is rapidly becoming standard in the field (e.g., the most recent NeurIPS tutorial on geometric deep learning; [1, 2, 3] to name a few). Our work builds directly on this to solve concrete architectural puzzles. To make this more accessible, we have overhauled the presentation and added extensive empirical validations
>
> ## Point 1: Intuition for geometric concepts
>
> We agree with your perspective. Originally, we included the Mahalanobis distance problem of contrastive learning as an intuitive example of why SSL is fundamentally a metric-learning problem. However, we have taken further steps to increase the accessibility of the work.
>
> We have added a new global diagram visually illustrating how the head folds high-variance augmentation orbits into a tight equivalence class, providing immediate visual intuition for "metric singularities." We also grounded the formalism with practical SSL context: defining augmented views as visually altered images that preserve semantics; the augmentation orbit as the manifold of all such variations (e.g., via rotation or color jitter); and the core goal of contrastive learning as mapping this entire orbit to a single point in space.
>
> Further, Figure 4 in the appendix may help provide some intuition of what compressing orbits does in some stylized, lower-dimensional view.
>
> ## Point 2: Empirical validation
>
> Please see our discussion in Point 3 for reviewer KAem. In short, we measured geometric properties of projection heads and backbones in a variety of real-life pretrained SOTA models. These include different architectures (CNN vs Transformer), different augmentations (e.g., cropping and rotation), and types of loss (contrastive and decorrelation).
>
> ## Point 3: ReLU gap
>
> Please see our discussion in Point 3 for reviewer jwre. In short, we ran ablations to show that the reason ReLU heads work is due to BatchNorm and careful selection of learning rates. This can explain why historically, SSL architectures using ReLU have been sensitive to the learning rate. If there is one practical takeaway from our work, it is that smooth, nonlinear activations (e.g., Swish/GELU) are vastly more robust to the choice of learning rate because they geometrically resolve the collapse traps that linear/ReLU heads fall into.
>
> ## Point 4: Why is this backbone/projection framework special?
>
> The reviewer is correct: a pullback metric applies at every layer. However, the backbone-head boundary is uniquely special due to the asymmetry of the objective and timescale separation of training.
>
> Consider a downstream task where we want to classify cat breeds by coat color. If we pretrain with a contrastive loss using color-jitter, we are forcing the network to become invariant to color. If we do not use a projection head, this color invariance is baked directly into the backbone, severely degrading its ability to perform the downstream color-based classification task.
> However, if we train with a projection head, the destruction of color information is pushed exclusively into the head because it "learns faster" (timescale separation). We can minimize the contrastive loss, then throw away the head. Because the backbone was shielded from throwing away the color information, it can still perform the downstream task.
>
> In short, the pretraining loss demands extreme geometric distortions (perfect local invariance and global whitening) that destroy linearly separable information. The network learns to push these destructive "metric singularities" exclusively into the projection head, which become the sacrificial layers it knows will not be probed downstream.
>
> As for why these are normally MLPs, they don’t need to be. We use MLPs in the proof of Theorem 3.2 to invoke the universal approximation theorem. Indeed, any type of layer can be used, but empirical experiments in e.g., SimCLRv2 found that shallow MLPs work well in practice. This geometric effect described above is the concise geometric motivation of why we throw away the projection head, which is the central question of the work.
>
> [1] "Understanding Contrastive Representation Learning through Alignment and Uniformity on the Hypersphere" (Wang & Isola, ICML 2020)
>
> [2] “Asymptotic and Finite-Time Guarantees for Langevin-Based Temperature Annealing in InfoNCE” (Chaudhry, OPT 2025)
>
> [3] “Understanding Dimensional Collapse in Contrastive Self-supervised Learning” (Jing et al., ICLR 2022)

---

> > ### Author Rebuttal · Reviewer_uX95 · 2026-04-03
> >
> > Thank you for the detailed response. The ReLU ablations (Reviewer jwre) meaningfully strengthen the paper. However, I have follow-up questions:
> >
> > 1. Point 4 in the authors' response above argues the head is necessary because the pretraining loss "demands extreme geometric distortions that destroy linearly separable information." But this frames the head as a patch for a fundamentally misaligned objective, i.e., the network must learn to not fully satisfy the loss it's optimizing. Does the Riemannian framework suggest what a loss would look like where invariance is enforced without inducing metric degeneracy in the backbone? Can we eliminate the need for a sacrificial buffer entirely? If not, what is the fundamental obstruction?
> > 2. Point 4 also relies heavily on timescale separation to explain why metric singularities are "pushed exclusively into the head." This assumption is load-bearing for the stability analysis, but the justification is qualitative ("upper layers generally move faster"). Did you verify that the head's parameters actually move faster than the backbone's in your experiments? Logging relative update norms per layer would be straightforward and would ground the assumption.
> > 3. Your depth ablation (1/2/3 layers, for Reviewer EDpS) is interesting but atheoretical. If the head is too deep, its lower layers would adapt at backbone-like speeds, weakening the timescale argument from Point 4. Does your framework predict an optimal depth, or only that some head is better than none?
> > 4. The conventional narrative attributes collapse avoidance to stop-gradients, EMA, predictor asymmetry, or BatchNorm, not the head's curvature. Your ReLU ablations show BatchNorm can substitute for smooth curvature. Is the mechanism in Theorem 4.1 necessary or merely one of several redundant mechanisms? A clearer statement of what your theory uniquely explains versus geometrically redescribes would help.
> > 5. On presentation: I appreciate the effort, but my original concern was structural rather than cosmetic. Without seeing the revised manuscript, I can't judge whether the changes address this. The paper as submitted reads as formalism-first with intuition appended, rather than intuition-driven with formalism supporting it. The references you cite (Wang & Isola; Jing et al.) are notably more accessible, which somewhat illustrates the gap.

---

> > > ### Author Response · Authors · 2026-04-07
> > >
> > > We thank the reviewer for these questions.
> > >
> > > 1.
> > > The reviewer raises a philosophical point: the pretraining loss is misaligned with downstream tasks. We agree, and this is the foundational compromise of SSL. The invariance objective is designed as a proxy and the loss function is not (fully) aligned with downstream tasks. The proxy is inherently destructive to certain information, but is useful as it allows us to extract semantically meaningful features without needing labelled data.
> > >
> > > Does our Riemannian framework suggest what a loss would look like where invariance is enforced without inducing metric degeneracy?  No: the buffer cannot be eliminated as long as the objective strictly enforces local invariance. The fundamental obstruction is that mapping a high-variance augmentation orbit to a single invariant point forces collapse. General-purpose utility and perfect invariance are mutually exclusive. As long as the loss demands invariance, metric degeneracy is unavoidable; the projection head (with timescale separation) simply confines that destruction to a disposable layer.
> > > If we wish to eliminate the projection head entirely, the loss must preserve the local metric tensor. We suggest two specific theoretical directions:
> > >
> > > -	Instead of strictly enforcing equality ($z_1 = z_2$), the loss enforces an invertible transformation. If view 2 is generated via augmentation $T$, the network is trained to apply the inverse in the latent space: $z_1 = T^{-1}(z_2)$. Because the latent space must support an invertible operation, it maintains full rank, preserving the local metric. This requires augmentations not to be lossy (e.g., a 90-degree rotation can be inverted, whereas a random crop cannot).
> > >
> > > -	Creating a loss that penalizes metric degeneracy by lower-bounding the singular values of the local Jacobian. While methods like VICReg enforce global full-rank covariance across a batch, a headless invariance loss would require a local variance regularizer applied directly to the individual augmentation orbit to prevent tangent space collapse.
> > >
> > > 2.
> > > We agree that empirical grounding of timescale separation strengthens the stability analysis. We conducted tracking runs during the early phase of training (the first 10 epochs). Following your suggestion, we logged the relative update magnitudes ($||\eta \nabla w||_2 / ||w||_2$) for both the projection head and backbone.
> > >
> > > We found that in the case of no BN, that timescale separation holds, with the head updates being >800 times faster than the backbone. Further, applying BN artificially constrains this relative update ratio to $\approx 1$, destroying natural timescale separation. We believe that this is a novel finding, and it reinforces our argument that BN acts as an artificial heuristic to force an escape from collapse, whereas intrinsic curvature natively provides both the geometry and the timescale separation required for stability. We are currently conducting a full 50-epoch run with tracking to produce the final figure/table for the revision. The ratios from our 10-epoch run can be viewed here: https://anonymous.4open.science/r/projection-head-geometry-1039/figures/timescale_seperation_ratios.png
> > >
> > > 3.
> > > The reviewer asks if our framework predicts an optimal depth. Our theory states: the optimal head is the shallowest network capable of universal approximation. A 1-layer linear head fails to adapt the metric. However, as the reviewer notes, a deep head violates timescale separation because its lower layers update at slower, backbone-like speeds. Thus, the head must achieve capacity via width and nonlinearity rather than depth. A sufficiently wide 2-3 layer MLP is a principled design, in line with the empirical guidelines.
> > >
> > > 4.
> > > The reviewer asks if intrinsic curvature is redundant with mechanisms like BN. Our theory is more fundamental in that it identifies curvature as the requirement for escaping collapse. Heuristics like ReLU+BN rely on discrete step-noise to artificially simulate this curvature. In contrast, nonlinear heads possess this curvature implicitly and are a principled way to resolve this problem due to our theorem.
> > >
> > > To clarify what we uniquely explain versus geometrically redescribe: we concede that framing dimensional collapse as (geometric) degeneracy reformalizes (information-theoretic) rank loss. However, our geometric framework uniquely solves two central SSL mysteries: 1) exactly why the projection head must be discarded, and 2) why noncontrastive losses do not collapse immediately (the head natively injects negative eigenvalues). To our knowledge, the mechanistic explanation for this latter phenomenon has remained an open question.
> > >
> > > 5.
> > > The reviewer makes a fair point. Our work targets complex mechanistic puzzles that only emerge once the foundational motivations of SSL (e.g., Wang & Isola) are established. However, we agree that this does not excuse taking intuition for granted. We refer to our final reply to reviewer EDpS for some concrete additions.

---

### Official Review · Reviewer_KAem · 2026-03-12

**Soundness:** 3
**Presentation:** 3
**Significance:** 2
**Originality:** 2
**Overall Recommendation:** 3
**Confidence:** 4

**Summary:**

This paper studies the geometry of the projection head in standard self-supervised models (e.g. SimSiam, BYOL, SimCLR) and demonstrates three main phenomena. First, it demonstrates that linear and non-linear heads can whiten the spectrum of the representation. Second, it proves that head curvature and architectural asymmetry in non-contrastive models (SimSiam, BYOL) induce collapse avoidance when smooth activation functions are used in the head. Finally, it shows that the projection head acts as a geometric information bottleneck for which directions corresponding to augmentations are collapsed after the head, hurting downstream performance. The analysis is performed for linear and non-linear heads, and the claims are validated empirically on small-scale examples.

**Compliance With Llm Reviewing Policy:**

Affirmed.

**Key Questions For Authors:**

This paper tackles an important problem in the SSL field: the necessity of adding a trainable projection head discarded at inference. The authors make interesting propositions to answer this question using geometric tools, but they make strong assumptions that are hard to meet in practice (e.g. smoothness of the projection head) and the empirical validation is too narrow in its current state. I would recommend using realistic backbones on larger-scale data and different losses to empirically strengthen the main claims of the paper.

**Limitations:**

Yes.

**Strengths And Weaknesses:**

**Weaknesses**

-	Existential results on the geometric preconditioning formed by the head in Theorem 3.1 and 3.2: as acknowledged by the authors, they only prove that linear or non-linear head have the capacity to whiten the representation spectrum but i) they never tested it in practice on real-life experiments and ii) they do not study the actual optimal solution geometry. Both points are crucial to validate the claims and give more credits to this theory. Additionally, it seems hard to decouple the form of the self-supervised loss and the whitening effect of the projection head. For instance, Barlow Twins, W-MSE or ViCReg explicitly imposes a whitening of the features while InfoNCE imposes a gaussian distribution on the latent space, which is tightly related to whitening. In these cases, adding a projection head would seem useless since the whitening is already performed.
-	Assumption 1 regarding the smoothness of the activation function in the projection head is unrealistic for most self-supervised models but it appears to be crucial in the current theory for Theorem 4.1 and the empirical evidence in Fig. 2.  I would like to know why ReLU activation works equally well than GELU in real-life scenarios and whether it also helps to avoid collapse in non-contrastive methods. The two hypotheses made in the discussion regarding implicit variance constraints and discrete dynamics SGD updates should be tested in practice.
-	Limited empirical validation: the claims on the geometrical conditioning effect is never tested in practice and the impact of the non-linearity in the projection head to avoid collapse is also underexplored. Several experiments would strengthen this work: run ablations study with different backbones (CNN, ViT) and smooth vs non-smooth heads, report eigen spectrum of the features before/after the head to measure the conditioning effect, compare different losses (contrastive/non-contrastive) and the effect on the representation geometry before/after the head, report Hessian/negative-curvature measurements near collapse for different architectural configurations.
-	Limited insights on invariance and information bottleneck: the authors show that the information quantity after the head is strictly inferior to the one before, and it is more invariant to data augmentation after the head than before. All these results have already been shown early-on in SimCLR and Theorem 5.1 and 5.4 does not add much insight into these phenomena. It also raises the question of which augmentations should be used, i.e. the ones the representation space (before head) should be invariant to in order to maximize downstream performances.

**Strengths**

-	This work tackles an important problem in SSL: the need of projection heads during training to achieve state-of-the-art performances.
-	It introduces a new geometric interpretation of projection heads using a Riemannian metric learning perspective.
-	The analysis includes the main self-supervised frameworks such as SimCLR, SimSiam, BYOL, Barlow Twins or VICReg.
-	The proposition that non-linear projection heads induce curvature that destabilize training around collapse points for non-contrastive methods is interesting and novel.

---

> ### Author Rebuttal · Authors · 2026-03-28
>
> We thank the reviewer for identifying the empirical gaps in our submission. We agree that grounding our theories in practical work is crucial. Below, we highlight existing appendix experiments and new validations addressing your requests.
>
> ## Point 1: Existential results and decoupling the whitening effect
>
> Regarding the existential results: our theoretical restriction to existential proofs stems from the core focus of the paper being differential geometry. Guarantees on finding these geometries requires transitioning into optimization dynamics and stochastic analysis (e.g., analyzing SGD vs. Adam trajectories, discrete vs. continuous flow), which appeals to a different theoretical framework. However, to demonstrate these results apply in practice, we added an experiment tracking the optimization geometry (minimum eigenvalue, variance, and residual gradients) empirically (see Point 3).
>
> Second, the reviewer raises an excellent point regarding the confounding effect of losses that explicitly impose whitening (e.g., Barlow Twins) or uniformity (InfoNCE). How do we measure the head's contribution if the loss already enforces it? To strictly decouple the geometric preconditioning of the head from the loss function, we tracked the covariance condition number ($\kappa$) during a purely non-contrastive run (SimSiam). Because this loss lacks negative samples or explicit decorrelation penalties, it has no native whitening mechanism. Yet, as we will discuss in Point 3, the activation of the Swish head dynamically drives the condition number down from $6 \times 10^5$ to $2 \times 10^5$. This empirically isolates the effect, proving the head can act as an active geometric preconditioner entirely independently of the loss.
>
> Finally, if the loss enforces whitening, why is the projection head necessary? Because imposing severe geometric constraints (perfect global whitening and local invariance) directly on the backbone degrades its natural, informative topology. The head acts as a geometric buffer. For example, if a model is pretrained with InfoNCE using color-jitter without a head, the backbone becomes entirely color-invariant, severely degrading downstream tasks relying on color. The head absorbs these extreme invariance/whitening demands at the output space, shielding the upstream backbone from metric degeneracy.
>
> ## Point 2: Nonsmooth activation functions (ReLU)
>
> Please see Point 3 of our response to Reviewer jwre. Our new ablations show ReLU's escape relies on BatchNorm and discrete step-size effects. This explains why SSL with ReLU is historically notoriously sensitive to hyperparameter choices.
>
> ## Point 3: Empirical validation on large-scale problems
>
> Regarding the request for broader validation, we would first like to gently point out that alongside our main ResNet-18 (CNN) experiments, our appendix already contains ablations using VIT. These ViT runs showed similar geometric qualities to the CNNs, leading us to conjecture that the dataset manifold, rather than the specific network architecture, primarily drives these metric adaptations. Furthermore, regarding large-scale validation, our appendix section "Analysis of Pretrained Backbones" already evaluates publicly available, large-scale checkpoints for DINO, VICReg, and Barlow Twins. This analysis spans different augmentations, CNN/ViT architectures, and contrastive/decorrelative losses.
>
> To address your request for Hessian measurements near collapse, we conducted a new experiment tracking the optimization geometry. First, we validated our structural asymmetry assumption: we logged the residual gradient norms near collapse and found they remain strictly bounded away from zero ($\approx 10^{-4}$) for nonlinear heads.
> Second, this tracking demonstrated the problem with linear heads: ReLU cannot escape a collapsed state because its Hessian's minimum eigenvalue remains exactly 0, leading to zero recovery in representation variance. Conversely, Swish successfully escapes because the head natively generates spikes of negative eigenvalues (negative curvature) that reshape the manifold.
>
> ## Point 4: Limited insights on invariance
>
> We fully agree that early empirical works (e.g., SimCLR) and recent information-theoretic analyses (e.g., Ouyang 2025) have observed and described the projection head's bottleneck effect. However, we believe our distinct Riemannian geometric perspective offers an elegant and mathematically precise formalization of how this occurs mechanistically. For example, viewing a decrease in rank explicitly as the loss of curvature along specific augmentation-tangent directions.
>
> Regarding which augmentations maximize downstream performance: our framework explains how the architecture absorbs invariance, but selecting optimal augmentations remains a dataset-specific question of inductive bias. Optimizing the augmentation policy is vital practically, but orthogonal to our theoretical analysis of the projection head's underlying geometry.

---

> > ### Author Rebuttal · Reviewer_KAem · 2026-04-01
> >
> > The hypothesis drawn about the necessity of the projection head when the loss already enforces whitening is related to point 4 and is, in my opinion, still unaddressed. I agree it is not an easy question, and it may be dataset-dependent (although the assumption behind foundation models suggests otherwise). However, the theoretical arguments are not strong enough to give a clearer explanation besides what is already known.

---

> > > ### Author Response · Authors · 2026-04-01
> > >
> > > We appreciate reviewer KAem pressing this point. It strikes at the heart of why explicit decorrelation losses (like VICReg) still fail without projection heads. The reviewer is correct that the community empirically knows the head acts as a 'buffer,' but our theoretical framework mathematically formalizes exactly _why_ the backbone cannot serve as its own buffer.
> > >
> > > ## Point A: Explaining why decorrelation losses require a head
> > >
> > > If a whitening/invariance loss is applied directly to the backbone $z$, Proposition 5.3 dictates that the metric singularity must occur within $z$ itself. Consequently, by Theorem 5.4, the Fisher Information Matrix of $z$ would lose rank along all augmentation directions. This annihilates the linearly separable information required for downstream tasks. The head is not just a heuristic buffer; its Jacobian $J_h$ is a necessity required to absorb the nullspace so the backbone's Fisher Information remains full rank.
> > >
> > > Furthermore, losses like VICReg create extreme geometric contradictions: the invariance term forces augmented views together, while the decorrelation term forces the covariance matrix to be full rank (pushing representations apart). This creates massive topological stress. As shown in Table 4, the projection head resolves this by taking on extreme curvature (e.g., VICReg rotation curvature jumps from 8.51 in the backbone to 18.28 in the head). By Proposition 3.3, linear maps or identity functions (i.e., no head) cannot resolve this curvature. The nonlinear head acts as a geometric 'crumple zone,' allowing the loss to achieve full-rank decorrelation in $h(z)$ while permitting the backbone $z$ to remain linearly clustered.
> > >
> > > ## Point B: Formalization is a step towards dynamics
> > >
> > > While the community has strong empirical heuristics about projection heads, heuristics are not proofs. Transitioning these phenomena from empirical observations to differential geometry provides a shared vocabulary for the geometric deep learning community. The pretraining objective is ultimately just a proxy for the downstream task. Because the ‘volume’ of the augmentation tangent space scales combinatorically with the number of applied transformations, relying on empirical guesswork to understand what is being filtered is insufficient, heavy handed, and only works on huge scales. Our geometric framework formally quantifies exactly what information is being compressed into the metric singularity.
> > >
> > > Finally, this posing the problem geometrically lays the groundwork for solving SSL's notorious hyperparameter sensitivity. As recent work has shown, the optimal choice of temperature/hyperparameters is highly data-dependent e.g., [1, 2, 3]. While we do not focus on the dynamics, our framework provides the theoretical foundation required to eventually derive data-aware hyperparameters under some choice of dynamics that can be imposed on training. We wish to highlight that the theory for the geometry is quite general, and for the dynamics is very specific to the choice of optimizer.
> > >
> > > [1] Temperature Schedules for Self-Supervised Contrastive Methods on Long-Tail Data, Kukleva et al.
> > >
> > > [2] Not All Semantics Are Created Equal: Contrastive Self‐Supervised Learning with Automatic Temperature Individualization,  Qiu et al.
> > >
> > > [3] Asymptotic and Finite-Time Guarantees for Langevin-Based Temperature Annealing in InfoNCE, Chaudhry.

---

### Decision · Program_Chairs · 2026-04-30

**Decision:**

Accept (regular)

**Comment:**

I recommend accepting the paper into the program.

The paper provides new theory on the role of projection heads in self-supervised learning. The theoretical contributions were positively noted by reviewers, with two major shared concerns:

First, the presentation and illustration of the results: Multiple reviewers noted that the paper is not readily accessible to a general ML audience in the way it is written currently. The theory is presented in an over-complicated way and the paper is quite light on toy examples and visualizations to communicate the main results to a broader audience. While the authors made some efforts to share better visualizations (https://anonymous.4open.science/r/projection-head-geometry-1039/figures/augementation_orbit_figure.png), a revision of the paper is required to fully meet the reviewer concerns.

Second, the depth of empirical validation: The main paper provides CIFAR scale results without clear (obvious) connection to the theory. Reviewers proposed several additional experiments and the authors responded with initial results.

The theoretical contribution was seen as novel and relevant and well positioned in the literature. Rejecting the paper based on limitations in clarity and experimental results alone seemed inappropriate for a venue like ICML. *However*, the authors should now make a serious attempt at enhancing the clarity of the paper, well beyond what they hinted at during the rebuttal. I believe that this will also strengthen the broader interest in the work once published. Additional (illustrative) empirical results from the rebuttal period should be added to the main paper as well. Finally, I would encourage the authors to cite additional related work. As it stands, the reference list is rather short.